# A human immune dysregulation syndrome characterized by severe hyperinflammation with a homozygous nonsense Roquin-1 mutation

S.J. Tavernier[1,2,3,23], V. Athanasopoulos[4,5,23], P. Verloo[6,23], G. Behrens[7,8], J. Staal [2,3], D.J. Bogaert[1,9], L. Naesens[1,10], M. De Bruyne [1,11], S. Van Gassen[12,13], E. Parthoens[14], J. Ellyard[4], J. Cappello[4], L.X. Morris[15], H. Van Gorp[10,16], G. Van Isterdael [3,17], Y. Saeys[12,13], M. Lamkanfi[10,16], P. Schelstraete[9], J. Dehoorne[18], V. Bordon[9], R. Van Coster[6], B.N. Lambrecht[19,20,21], B. Menten[11], R. Beyaert [2,3], C.G. Vinuesa[4,5], V. Heissmeyer[7,8], M. Dullaers [1,22] & F. Haerynck[1,9]*

Hyperinflammatory syndromes are life-threatening disorders caused by overzealous immune cell activation and cytokine release, often resulting from defects in negative feedback mechanisms. In the quintessential hyperinflammatory syndrome familial hemophagocytic lymphohistiocytosis (HLH), inborn errors of cytotoxicity result in effector cell accumulation, immune dysregulation and, if untreated, tissue damage and death. Here, we describe a human case with a homozygous nonsense R688* *RC3H1* mutation suffering from hyperinflammation, presenting as relapsing HLH. *RC3H1* encodes Roquin-1, a posttranscriptional repressor of immune-regulatory proteins such as ICOS, OX40 and TNF. Comparing the R688* variant with the murine M199R variant reveals a phenotypic resemblance, both in immune cell activation, hypercytokinemia and disease development. Mechanistically, R688* Roquin-1 fails to localize to P-bodies and interact with the CCR4-NOT deadenylation complex, impeding mRNA decay and dysregulating cytokine production. The results from this unique case suggest that impaired Roquin-1 function provokes hyperinflammation by a failure to quench immune activation.

[1] Primary Immune Deficiency Research Lab, Department of Internal Medicine and Pediatrics, Centre for Primary Immunodeficiency Ghent, Jeffrey Modell Diagnosis and Research Centre, Ghent University Hospital, Ghent, Belgium. [2] VIB Center for Inflammation Research, Unit of Molecular Signal Transduction in Inflammation, Ghent, Belgium. [3] Department of Biomedical Molecular Biology, Ghent University, Ghent, Belgium. [4] Department of Immunology and Infectious Disease and Center for Personalised Immunology (NHMRC Centre for Research Excellence), John Curtin School of Medical Research, Australian National University, Canberra, Australia. [5] Centre for Personalised Immunology (CACPI), Shanghai Renji Hospital, Shanghai Jiao Tong University, Shanghai, China. [6] Department of Internal Medicine and Pediatrics, Division of Pediatric Neurology and Metabolism, Ghent University Hospital, Ghent, Belgium. [7] Institute for Immunology, Biomedical Center, Ludwig-Maximilians-Universität München, Planegg-Martinsried, Germany. [8] Research Unit Molecular Immune Regulation, Helmholtz Zentrum München, Munich, Germany. [9] Department of Internal Medicine and Pediatrics, Division of Pediatric Immunology and Pulmonology, Ghent University Hospital, Ghent, Belgium. [10] Department of Internal Medicine and Pediatrics, Ghent University Hospital, Ghent, Belgium. [11] Center for Medical Genetics, Ghent University Hospital, Ghent, Belgium. [12] VIB Center for Inflammation Research, Unit of Data Mining and Modeling for Biomedicine, Ghent, Belgium. [13] Department of Applied Mathematics, Computer Science and Statistics, Ghent University, Gent, Belgium. [14] VIB Bioimaging Core, VIB Center for Inflammation Research, Ghent, Belgium. [15] The Australian Phenomics Facility, John Curtin School of Medical Research, Australian National University, Canberra, Australia. [16] VIB Center for Inflammation Research, Ghent, Belgium. [17] VIB Flow Core, VIB Center for Inflammation Research, Ghent, Belgium. [18] Department of Internal Medicine and Pediatrics, Division of Pediatric Rheumatology, Ghent University Hospital, Ghent, Belgium. [19] Department of Internal Medicine and Pediatrics, Division of Pulmonology, Ghent University Hospital, Ghent, Belgium. [20] VIB Center for Inflammation Research, Unit for Immunoregulation and Mucosal Immunology, Ghent, Belgium. [21] Department of Pulmonary Medicine, ErasmusMC, Rotterdam, The Netherlands. [22] Ablynx, a Sanofi Company, Zwijnaarde, Belgium. [23] These authors contributed equally: S. J. Tavernier, V. Athanasopoulos, P. Verloo. *email: Filomeen.Haerynck@ugent.be

Hyperinflammatory syndromes are life-threatening disorders caused by severe and uncontrolled immune cell activation and hypercytokinemia. These syndromes comprise a constellation of distinct entities such as hemophagocytic lymphohistiocytosis (HLH), macrophage activation syndrome, sepsis and the cytokine release syndrome in the setting of immunotherapy. The clinical presentation shares a number of features such as unremitting fever, splenomegaly, coagulopathy, hepatitis, cytopenia and, if unrestrained, multi-organ failure and death. At the heart of these diseases lies an uncontrolled immune response to a persisting trigger, which can be pathogen driven or innocuous (self) antigen derived[1–4].

Especially in familial HLH (FHL), progress has been made to identify the underlying disease-causing genes. These variants are mostly situated in pathways that regulate cytotoxic granule function (e.g., *PRF1*) or exocytosis (e.g., *RAB27A, LYST*). In these conditions, HLH is often the only manifestation of disease but can also be part of a broader syndrome[2]. Additional inborn errors of the immune system such as X-linked lymphoproliferative disease (*SH2D1A, XIAP*) are prone to the development of HLH[2]. Although these hyperinflammatory episodes in FHL occur typically in the first years of life, hypomorphic mutations of these genes can give rise to atypical HLH at adult age[5,6]. Currently, hematopoietic stem cell transplantation is considered to be the only curative treatment option in FHL[7].

Roquin-1, encoded by *RC3H1*, recognizes and binds to RNA by the virtue of its ROQ domain and the adjacent C3H1 zinc finger[8–14]. It acts as a post-transcriptional regulator that typically promotes mRNA degradation[15] but also protein translation inhibition has been reported[16]. As such, it controls immune-relevant proteins such as ICOS, OX40, CTLA4, REL, IκBδ, IκBζ, and TNF among others[9,17–19]. Roquin-1 has no intrinsic nuclease activity but relies on the recruitment of RNA decapping and deadenylation complexes[9,17,20]. Furthermore, Roquin-1 regulates RNA expression in cooperation with the endonuclease Regnase-1, relying on the binding of RNA by the Roquin ROQ domain and the nuclease activity of Regnase-1, although also spatiotemporal distinct modes of action of these regulators have been suggested[19,21]. As a consequence of its function, Roquin-1 can colocalize with P-bodies, cytoplasmic regions in which stalled mRNA storage and post-transcriptional regulation occurs[22].

Roquin-1 came under the immunological limelight with the original description of sanroque mice by Vinuesa and Goodnow[23]. The sanroque mouse strain, carrying a homozygous point mutation (M199R) in the ROQ domain of *Rc3h1*, was the result of an ethylnitrosourea mutagenesis screen to identify repressors of autoimmune responses. These mice acquired a lupus-like disease with anti-nuclear antibodies, splenomegaly and lymphadenopathy, became anemic, thrombocytopenic and developed hepatitis and glomerulonephritis. The underlying immune dysregulation was characterized by accumulation of T follicular helper (Tfh) cells and germinal center (GC) B cells[23,24]. Subsequent reports revealed that in the presence of the hypomorphic M199R variant, ICOS expression and interferon-γ release increased, promoting Tfh cell proliferation and impairing the negative selection of autoimmune GC B cells[15,25].

The immunoregulatory function of Roquin-1 was further unraveled making use of immune cell specific conditional knockout mice. Loss of Roquin-1 in T cells or B cells resulted in effector T cell expansion, eosinophilia and monocytosis but failed to induce Tfh cell and GC B cell accumulation[26]. The generation of mice lacking both Roquin-1 and Roquin-2 revealed functional redundancy as loss of both paralogs aggravated immune dysregulation and prompted Tfh cell and GC B cell expansion[18]. These findings reveal the complex regulation and crucial role of Roquin-1 in the murine immune system.

Here, we describe a hyperinflammatory syndrome presenting as relapsing HLH in a patient with a homozygous nonsense mutation (R688*) in *RC3H1*, yielding a truncated Roquin-1. In-depth immunophenotyping reveals pronounced immune dysregulation bearing striking resemblance with the phenotype observed in Roquin-1 mouse models. By detailed analysis of the sanroque mice, we unveil additional parallels between human and murine disease. Inhibition of JAK1/2 signaling in the sanroque mice mitigates disease. Mechanistically, the truncated R688* Roquin-1 does not colocalize with P-bodies, fails to interact with the CCR4-CNOT1 deadenylation complex and delays the decay of the Roquin-1 target *ICOS* mRNA. Transduction of the *Rc3h1* mutants in murine T cells deficient for Roquin-1 and -2 reveals a pronounced impairment of the truncated Roquin-1 to reconstitute repression of known targets such as ICOS, Ox40 and CTLA4. Furthermore, these experiments indicate that the R688* variant fails to control the production of a number of cytokines such as TNF, IL-2 and IL-17A. In conclusion, our work highlights that post-transcriptional control by Roquin-1 is critical in the regulation of the human immune system.

## Results

**Identification of a homozygous nonsense R688\* RC3H1 variant.** We performed whole exome sequencing (WES) to identify causal mutations in the case of an 18-year-old male, who was referred to our center at age 11 suffering from hyperinflammation clinically resembling hemophagocytic lymphohistiocytosis (HLH) (Table 1). The patient was treated according to the HLH-2004 protocol[27]. After termination of Cyclosporin A (CSA), at age 13, disease reactivation was observed, and clinical course only

| **Table 1 Characteristics of relapsing hyperinflammatory syndrome in the R688\* patient** | | |
|---|---|---|
| **Characteristics** | | |
| | Episode 1 | Episode 2 |
| Age | 11 years | 13 years |
| *Clinical manifestations* | | |
| Fever (*T* > 38 °C) | >4 weeks | >2 weeks |
| Splenomegaly | Mild | Prominent |
| Hepatomegaly | Mild | Prominent |
| Lymphadenopathy | Present | Present |
| *Biochemical features* | | |
| Hemoglobin (g/dL) | **6.2** (11.5–15.5) | **9.9** (13–16) |
| Platelets ($10^3$/μL) | **42** (156–408) | 234 (156–408) |
| Leukocytes ($10^3$/μL) | **2.13** (4.5–12) | 5.57 (4.5–12) |
| Neutrophils (cells/μL) | **1299** (2500–8000) | 3130 (2500–8000) |
| Monocytes (cells/μL) | **50** (500–1000) | **260** (500–1000) |
| Lymphocytes (cells/μL) | **809** (1500–6500) | 1700 (1500–6500) |
| Ferritin (ng/μL) | **35199** (7–142) | **5162** (7–142) |
| Fibrinogen (mg/dL) | **<60** (200–400) | 305 (200–400) |
| Triglycerides (mg/dL) | **870** (32–125) | **996** (32–125) |
| Soluble CD25 (pg/mL) | NA | **16944** (632–5054) |
| *Features of hemophagocytosis* | | |
| Bone marrow aspirate | Mild | NA |
| *NK-cell activity* | | |
| Target cell killing | NA | Normal |
| CD107a expression | NA | Normal |
| *Additional features* | | |
| Gamma-GT (U/L) | **908** (3–22) | **274** (2–42) |
| AST (U/L) | **1482** (11–50) | **209** (0–37) |
| ALT (U/L) | **199** (7–40) | **168** (7–40) |

Units of measurements are mentioned in parentheses, bold characters indicate values below or above normal range. Normal ranges are indicated in parentheses

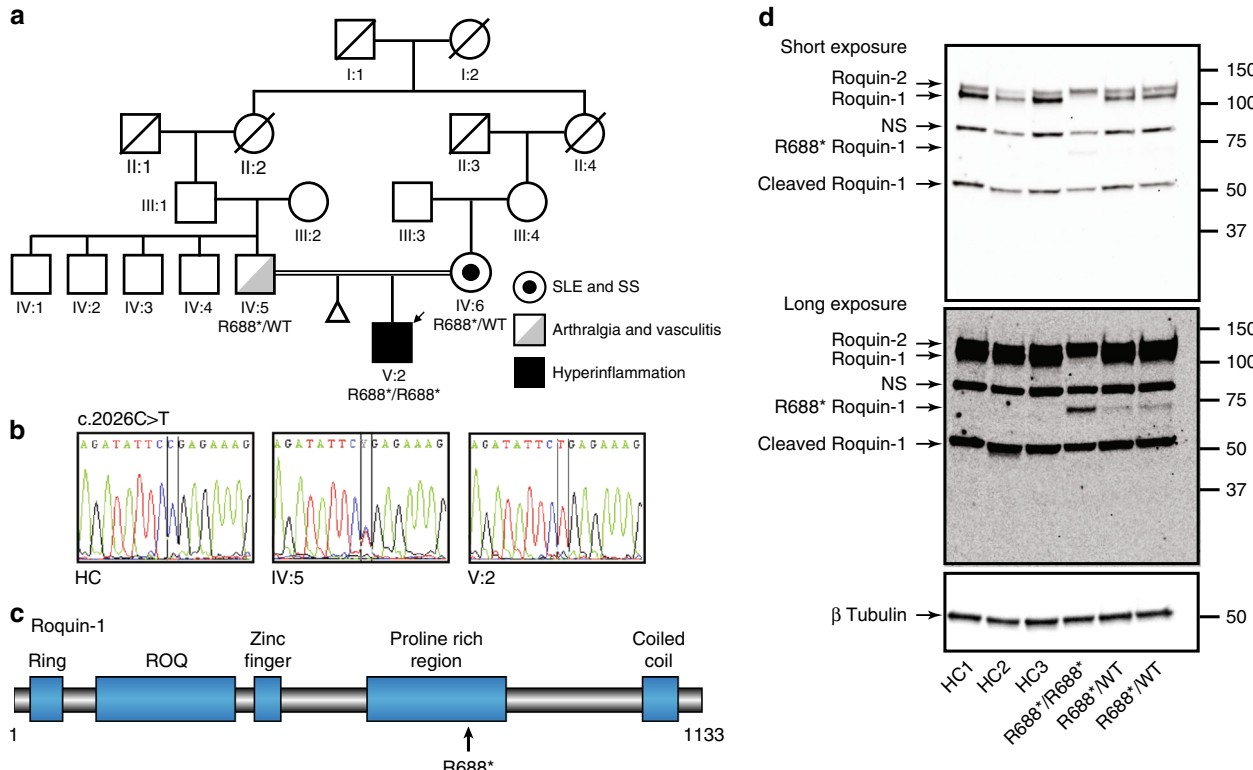

**Fig. 1** Identification of a nonsense R688* mutation in *RC3H1* in a consanguineous family. **a** Family pedigree indicating the index patient (V:2) with an arrow, the consanguineous link (double line) between the index patient's parents and reported medical conditions as indicated in the legend. **b** Sanger sequencing of complementary DNA from selected individuals and control. **c** Graphical representation of Roquin-1 protein structure with indication of the R688* mutation. RING: Really Interesting New Gene zinc finger motif. ROQ: roquin-family RNA binding domain. Zinc finger: CCCH zinc finger motif. Coiled Coil: Coiled coil domain. **d** Immunoblot analysis of Roquin-1, its paralog Roquin-2, their cleavage products and the truncated R688* mutant in healthy controls (HC), the R688* proband and both parents. β-Tubulin is used as a loading control. NS: nonspecific band, SLE: systemic lupus erythematosus, SS: Sjögren's syndrome. Source data are provided as a Source Data file

ameliorated under treatment with CSA (Table 1). No infectious agent or autoimmune trigger could be identified (Supplementary Fig. 1A–C). Despite good clinical control, laboratory findings revealed ongoing inflammation under CSA treatment (Supplementary Fig. 1D–G). Furthermore, the patient suffers from chronic hepatitis and dyslipidemia (Supplementary Fig 1H–J). This immune dysregulation syndrome developed on top of a dysmorphic phenotype (short stature, webbed neck) and mild mental retardation. The patient is the first child of Belgian consanguineous parents with Spanish roots. Family history reveals a spontaneous abortion of the first pregnancy and a predisposition to autoimmune mediated pathology (Fig. 1a).

We were unable to identify pathogenic variants in known HLH genes nor in any other described PID gene (Supplementary Table 1). Immunological work-up showed normal NK-cell cytotoxicity, expression of perforin and CD107a and normal iNKT cell numbers, providing additional arguments against most familial HLH (Table 1 and ref. [28]). Ultimately, selection of variants predicted to result in a missense, nonsense, indel, or splice-site mutation uncovered a homozygous nonsense mutation in the *RC3H1* gene encoding Roquin-1: g.173931003G>A (ENST00000258349.4: c.2062C>T, ENSP00000258349.4: p. R688*) with pathogenic in silico predictions (CADD score = 40). Interrogation of public databases (dbSNP, gnomAD, ESP, Bravo) revealed that this R688* Roquin-1 variant has not yet been described in human populations[29]. Sanger sequencing confirmed the mutation located in exon 12, a region coding for a proline-rich domain in Roquin-1 (Fig. 1b, c). Both parents are heterozygous carriers of the mutation (Fig. 1a, b).

Whereas full-length Roquin-1 was undetectable in the case of the patient, longer exposure revealed a faster running protein at 75 kDa (Fig. 1d). Roquin-1 is cleaved by the paracaspase MALT1 upon TCR stimulation at R510 and R597[19]. Indeed, stimulation of patient-derived T cells with ionomycin and the phorbol ester PMA promoted the disappearance of this faster running protein. Pretreatment with the MALT1 inhibitor mepazine blocks Roquin-1 cleavage and confirmed the identity of the faster running protein (Supplementary Fig. 1K). In conclusion, we identified a homozygous nonsense R688* mutation in *RC3H1* encoding a truncated Roquin-1 in a patient with relapsing hyperinflammatory episodes.

**Immune dysregulation in the presence of the R688* RC3H1 variant**. We performed in-depth immunological phenotyping of the patient's peripheral blood mononuclear cells (PBMCs) to characterize the immunological abnormalities associated with the nonsense R688* *RC3H1* mutation. We analyzed this data using the unsupervised clustering and visualization algorithm Flow-SOM[30]. Through the use of a self-organizing map (SOM), FlowSOM assigns cells to a number of nodes and subsequently structures these nodes in a minimal spanning tree based on the expression of distinct markers. After identifying viable cells, the datafiles of the R688*/R688* patient and age-matched healthy controls (HCs) were concatenated into 1 dataset to generate a single FlowSOM tree for all individuals (Fig. 2a). FlowSOM was able to identify and cluster relevant immune cell populations and organize them in a coherent manner (Fig. 2a). We analyzed the

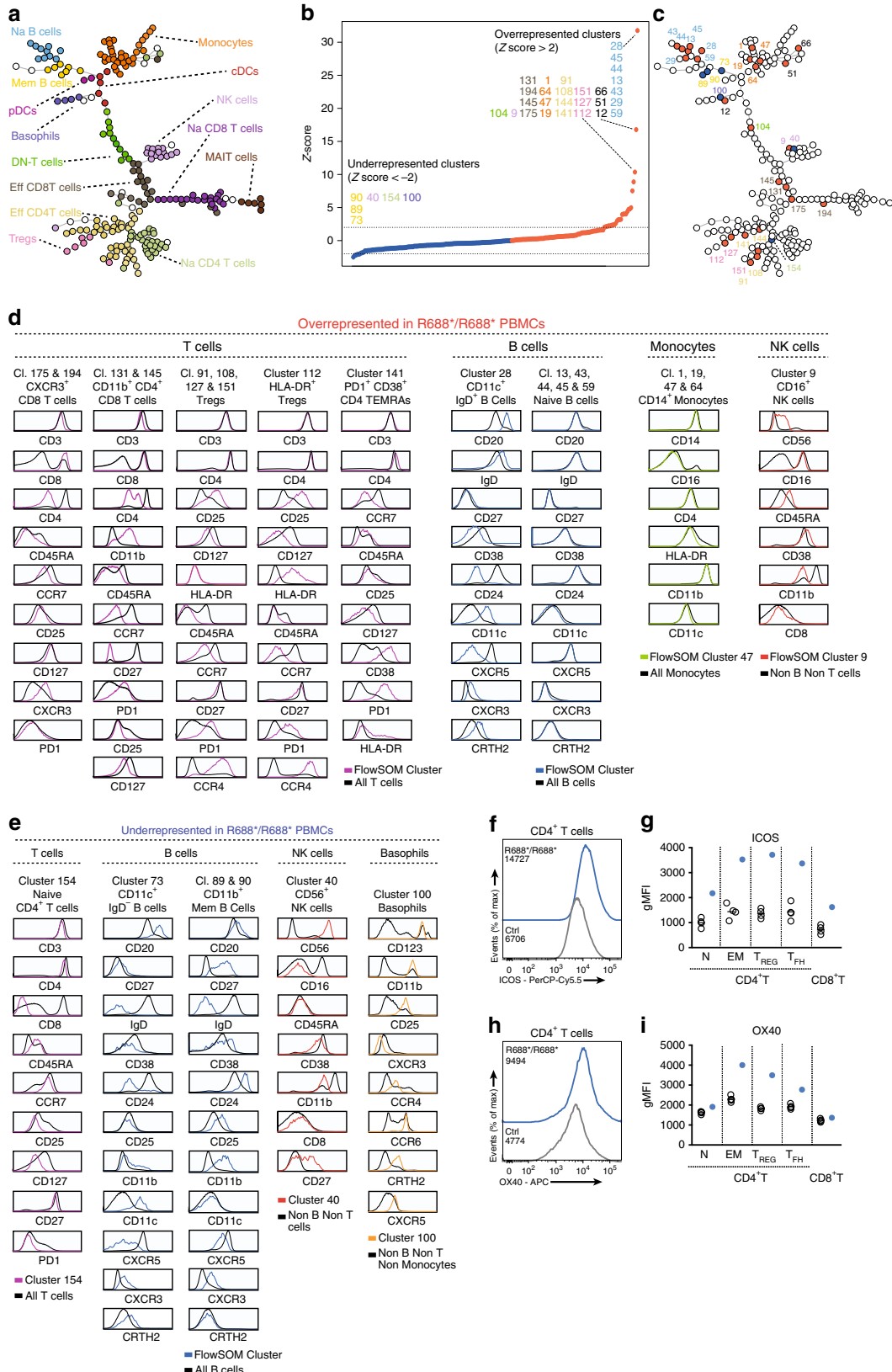

contribution of R688*/R688* immune cells in each node (Supplementary Fig. 2A) and identified nodes in which R688*/R688* immune cells were significantly under- or overrepresented (Z-score < −2 or >2) (Fig. 2b). By plotting these nodes onto the trained FlowSOM tree, we found that clusters containing naive B cells, CD14+ monocytes, effector CD4+ and CD8+ T cells,

regulatory T cells (Tregs) and CD16+ NK cells were overrepresented in the R688*/R688* patient whereas clusters identified as memory B cells, basophils, naive CD4+ T cells and CD56+ NK cells were underrepresented (Fig. 2c). An additional three clusters containing cellular debris (cluster 12) or doublet cells (clusters 51 and 66) appeared overrepresented in the

**Fig. 2** Analysis of the R688* proband peripheral blood mononuclear cells (PBMCs) reveals immune dysregulation. **a** FlowSOM tree of concatenated 29-parameter cytometry data of PBMCs obtained from seven HCs and the R688* proband. **b** Normalized data of the relative contribution of R688* proband PBMCs to each immune cell cluster. Percentage of R688* immune cells was normalized into a Z score based on HC mean and SD. Each cluster with a Z score > 2 (red) or <−2 (blue) was considered as a relevant immune cell population. Color of cluster number corresponds with panel **a**. **c** Clusters with a Z score > 2 (red) or <−2 (blue) were plotted onto FlowSOM tree. Color of cluster number corresponds with panel **a**. **d, e** Phenotypic description of overrepresented (**d**) and underrepresented (**e**) clusters in the R688* proband. Histograms depict expression profile of surface markers of given clusters (colored) compared with relevant immune cell populations (black). **f** Histogram representing ICOS expression on CD4+ T cells of a HC and R688* proband. Mean fluorescence is given. **g** Scatter dot plot of geometric mean fluorescence (gMFI) of ICOS in T cell subsets of HCs ($n = 4$) or proband. N: naive; EM: effector memory; Tfh: T follicular helper cell. **h** Histogram of OX40 expression on CD4+ T cells of a representative HC and R688* proband. Mean fluorescence is given. **i** Scatter dot plot of gMFI of OX40 in T cell subsets of HCs ($n = 4$) or proband. Data shown in (**a-i**) are representative for two independent experiments. Source data are provided as a Source Data file

R688*/R688* PBMCs (Supplementary Fig. 2A, C). These results corroborated to a large extent the classical supervised analyses performed on PBMCs collected at different ages (Supplementary Fig. 2B).

**Chronic activation and exhaustion of R688*/R688* T and B cells.** The phenotype of the cell clusters was further refined by analyzing surface marker expression (Fig. 2d, e). The over-represented effector CD8+ T cell clusters contained both CXCR3+ T cells (clusters 175 and 194) and CD11b+ CD27− PD1+ CD8+ T cells with variable expression of CD4 (clusters 131 and 145). The latter T cell population is also observed during viral infections and autoimmune diseases and represents an exhausted population with cytotoxic capacity (Fig. 2d and refs. [31,32]). Within the overrepresented clusters annotated as Tregs and effector CD4+ T cells, we identified a large number of clusters compatible with bona fide Tregs (clusters 91, 108, 112, 127, and 151) (Fig. 2d). Among these, cluster 112 contained activated effector Tregs (HLA-DR+) with highly suppressive capacity (Fig. 2d and ref. [33]). An additional population of CD4+ terminal effector memory T cells (TEMRAs) with elevated expression of the inhibitory molecule PD-1 (cluster 141) was similarly increased in the R688*/R688* PBMCs (Fig. 2d). Reflectory, 1 cluster (cluster 154) containing naive CD38lo CD4+ T cells appeared under-represented in the R688*/R688* PBMCs although manual gating could not identify reductions in naive CD4+ T cells (Fig. 2e and Supplementary Fig. 2B). Functional analyses were in line with these findings; intracellular cytokine staining demonstrated that both IL-17A+ CD4 T cells and IFNγ+ CD8 T cells were expanded (Supplementary Fig. 2B).

Among the cell clusters with the highest Z-scores, a population of B cells with a distinct surface marker expression could be identified (cluster 28, Fig. 2b, d). This CD20hi CD11c+ CD24− CD27− CD38− population expands during chronic inflammation and has been observed in a number of autoimmune conditions including systemic lupus erythematosus (SLE), primary Sjögren's syndrome and common variable immunodeficiency (CVID)[34]. This population lacked the chemokine receptor CXCR5, necessary for trafficking to B cell zones in secondary lymphoid organs but rather expressed CXCR3 and CRTH2, suggesting that these cells might migrate to sites of inflammation. This B cell subset still expressed surface IgD, indicative of an unswitched phenotype (Fig. 2d). Similarly, naive B cells (clusters 13, 43, 44, 45, 59) were strongly increased in the R688*/R688* PBMCs whereas memory B cells (clusters 73, 89, 90) were reduced (Fig. 2d, e). Aside of a minor decrease in IgG2 levels, this does not lead to major defects in humoral immune responses (Supplementary Fig. 1A, Supplementary Fig. 2D, E). Analysis of specific polysaccharide antibody responses was not performed as additional vaccinations were refused.

**Increased expression of ICOS and OX40 by R688*/R688* T cells.** In mice, loss of post-transcriptional regulation by Roquin-1 results in increased expression of ICOS and Ox40 in T cells (Supplementary Table 2 and ref. [18]). Likewise, T cells of the R688*/R688* patient displayed augmented levels of both ICOS and OX40 (Fig. 2f–i). Careful comparison with the published findings on Roquin-1 (and Roquin-2) mouse models demonstrated additional analogies (Supplementary Table 2). In the absence of Roquin-1, mice develop a similar immunopathology characterized by the expansion of effector T cells and Tregs, monocytosis and eosinophilia. In contrast to the sanroque model and in mice in which T cells are deficient for both Roquin-1 and 2, the R688* Roquin-1 mutation did not result in the expansion of the CXCR5+ circulating counterparts of T follicular helper cells (cTfh) (Supplementary Table 2 and Supplementary Fig. 2B). In conclusion, our R688*/R688* PBMC phenotyping experiments revealed pronounced immune dysregulation which shared resemblance with Roquin mouse models.

**Adaptive and innate immunity contribute to hypercytokinemia.** The observed immune dysregulation is not sufficient to explain the hyperinflammatory episodes, which are the consequence of excessive T cell and/or monocyte/macrophage activation and uncontrolled cytokine release. As Roquin-1 is known to regulate the expression of proinflammatory cytokines such as TNF[17], we measured serum cytokines and found increases of both proinflammatory cytokines TNF, IL-1β, IL-6, IL-17A, IL-18, IFNγ, CXCL9, and regulatory mediators such as IL-1RA and IL-10 (Fig. 3a). This hypercytokinemia was observed under sustained CSA treatment, indicating that other immune cells in addition to T cells contribute to the observed hypercytokinemia. Hemophagocytic lymphohistiocytosis (HLH) and macrophage activation syndrome (MAS) represent two distinct entities and recent studies have demonstrated that IL-18 and CXCL9 might serve as valuable biomarkers to distinguish both[35]. Here, analysis of IL-18 and CXCL9 concentrations suggested that the observed immune dysregulation was more akin to HLH than MAS (Fig. 3a and Supplementary Fig. 3A).

To study the contribution of adaptive and innate immune cells, T cells and monocytes were enriched from PBMCs and stimulated ex vivo. TNF and IFNγ were increased in the supernatant of PMA and ionomycin stimulated T cells (Fig. 3b, c). Monocytes were stimulated with both ATP and LPS to assess the activation of the inflammasome in the presence of the R688* variant. Although LPS induced a higher secretion of TNF and IL-6 by R688*/R688* monocytes, IL-1β and IL-18 release was similar to HCs (Fig. 3d and Supplementary Fig. 3B). These results indicated that both innate and adaptive immune cells contribute to disease (Fig. 3b–d).

Roquin-1 exerts control over immune responses by virtue of its post-transcriptional regulation of RNA[36]. Indeed, mRNA transcripts of established targets were upregulated in the R688*/R688*

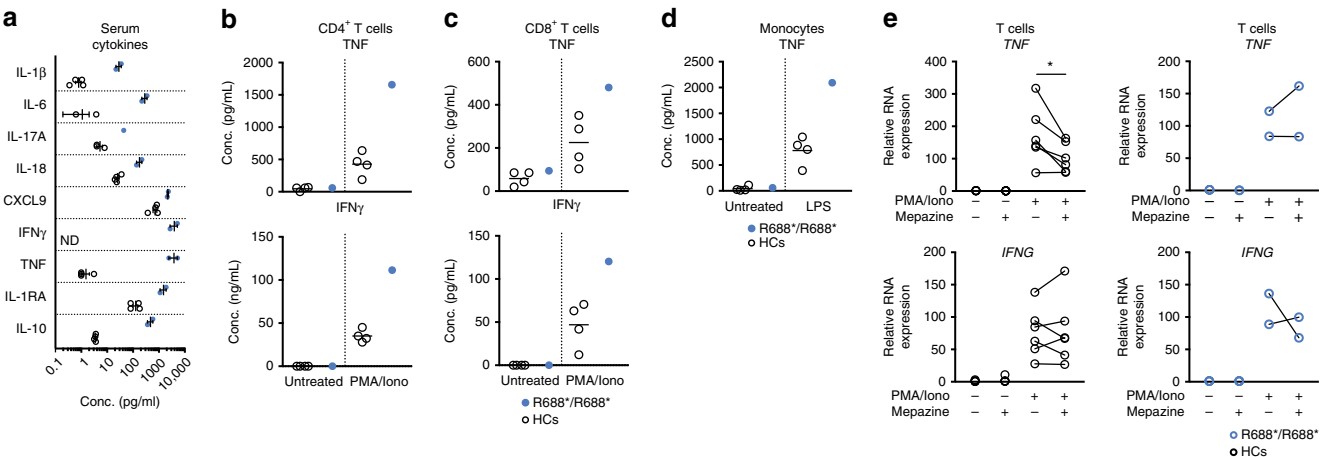

**Fig. 3** T cells and monocytes contribute to hypercytokinemia in the R688*/R688* proband. **a** Serum concentration of the cytokines IL-1β, IL-1RA, IL-6, IL-10, IL-17A, IL-18, IFNγ, CXCL9, and TNF in HCs (n = 4) and proband (two biological replicates) or HCs (n = 3) and proband (one biological replicate) in the case of the cytokine IL-17A. Mean and SEM are depicted. **b**, **c** ELISA of TNF and IFNγ produced by in vitro PMA/ionomycin stimulated CD4+ T cells (**b**) or CD8+ T cells (**c**) of HCs (n = 4) and proband. **d** ELISA of TNF produced by monocytes of HCs (N = 4) or proband treated in vitro overnight with LPS. **e** RT-qPCR quantifying TNF and IFNG transcripts in PMA/ionomycin stimulated T cells in absence or presence of mepazine pretreatment (20'). Cells were sampled 1 h after stimulation. Data was normalized using the housekeeping genes HPRT and GAPDH. HCs (n = 6). *p < 0.05 (paired t-test). R688* proband (n = 2). Data shown are accumulated from two independent experiments (**a**, **e**) or representative for two independent experiments (**b–d**). Source data are provided as a Source Data file

T cells (Supplementary Fig. 3C). To study the effects of truncated R688* Roquin-1 on mRNA transcripts of TNF and IFNG in more detail, we stimulated R688* or HC T cells with or without pretreatment of the cells with mepazine (Fig. 3e). Mepazine inhibits Roquin-1 degradation upon T cell activation (Supplementary Fig. 1K) and promotes Roquin-1 dependent mRNA regulation. Confirming the impaired function of R688* Roquin-1, TNF mRNA was not reduced in the setting of stimulated R688*/R688* T cells pretreated with mepazine (Fig. 3e). In contrast, IFNG expression was not decreased by pretreatment with mepazine in HC T cells (Fig. 3e).

**Sanroque mice suffer from systemic hyperinflammation.** To study immune dysregulation in the presence of impaired Roquin-1 function in more detail, we made use of sanroque mice. The M199R variant acts as a hypomorphic allele but does not result in postnatal lethality as observed in Roquin-1 null mutants, rendering this strain ideally suited for analysis[23,26]. Similar to the R688* mutation, sanroque mice endured pronounced hypercytokinemia, illustrated by the increased concentrations of IL-2, IL-6, IL-10, IFNγ, CXCL9, and TNF (Fig. 4a). An unchanged IL-17A concentration was noted and is reminiscent of the differences between sanroque mice and the Rc3h1-2^fl/fl; CD4-Cre mice, in which Th17 differentiation increased similar to what we observe in the Roquin-1 R688*/R688* patient[19]. In contrast, the observed hypercytokinemia in sanroque mice did not result in full-blown hyperinflammation resembling HLH. Whereas sanroque mice developed pronounced splenomegaly, mild thrombocytopenia, tendency to anemia, increased soluble CD25 (sCD25) and hepatitis, other hallmarks of HLH such as neutropenia, hyperferritinemia and increased triglycerides were absent (ref. [23], Fig. 4b–e and Supplementary Fig. 4A–D).

**Sanroque mice develop severe disease upon CpG injection.** Transplantation of sanroque bone marrow cells into sublethal irradiated CD45.1 mice recapitulated main features of the immune dysregulation such as ICOS upregulation and splenomegaly. Disease progression was observed with progressive leukopenia and anemia (Supplementary Fig. 4E-H). Spleen

immunophenotyping revealed a decrease in B cells without maturation defects and infiltration with both granulocytes and monocytes (Fig. 4f and Supplementary Fig. 4J). As reported in[25], Tregs and Tfh cells were increased and both CD4+ and CD8+ T cells displayed an effector memory phenotype (Fig. 4g, h). Liver analysis revealed pronounced tissue infiltration by monocytes (Fig. 4i). These data confirm that impaired Roquin-1 function in hematopoietic cells is sufficient to induce immune dysregulation. This systemic inflammation in sanroque chimeras might render these mice more sensitive to the occurrence of hyperinflammatory disease. To test this hypothesis, mice were subjected to CpG injections every 2 days, a known macrophage activation syndrome (MAS) model[37,38]. Repetitive CpG ODN-1826 injections uniformly resulted in splenomegaly and cytopenia, in sanroque and control chimeric mice (Supplementary Fig. 4K). Careful analysis revealed that the sanroque chimeric mice produced more TNF and IL-10 and lost more weight upon CpG injection compared with control mice (Fig. 4j, k). These results indicate that reduced Roquin-1 function in sanroque mice results in a more pronounced hyperinflammation.

**Cell intrinsic and extrinsic effects of sanroque mutation.** The crucial role of uncontrolled cytokine release in the phenotype of sanroque mice was highlighted by the Ifngr−/− sanroque mice[25]. Loss of IFNγ signaling reduced splenic hypercellularity, Tfh and GC B cells numbers and ameliorated autoimmunity[25]. To study the influence of hypercytokinemia in sanroque mice in more detail, sublethal irradiated CD45.1/2 mice were transplanted with 30/70 mixed CD45.2 sanroque and CD45.1 wild-type (WT) bone marrow (BM) cells. Chimeras generated with 30/70 mixed CD45.2 WT/CD45.1 WT BM cells functioned as controls. Analysis revealed a cell intrinsic expansion of CD4+ T cells (Fig. 5a). This was associated with a high percentage of CD4+ effector memory (EM) T cells expressing increased levels of ICOS and a marked differentiation into Tregs (Fig. 5b, c and Supplementary Fig. 5A). Similarly, we observed a cell intrinsic maturation into CD8+ EM T cells (Fig. 5d, e). The CD45.2 sanroque/CD45.1 WT BM chimeras recapitulated the reduced number of B cells and delayed maturation of NK cells, highlighting cell intrinsic effects

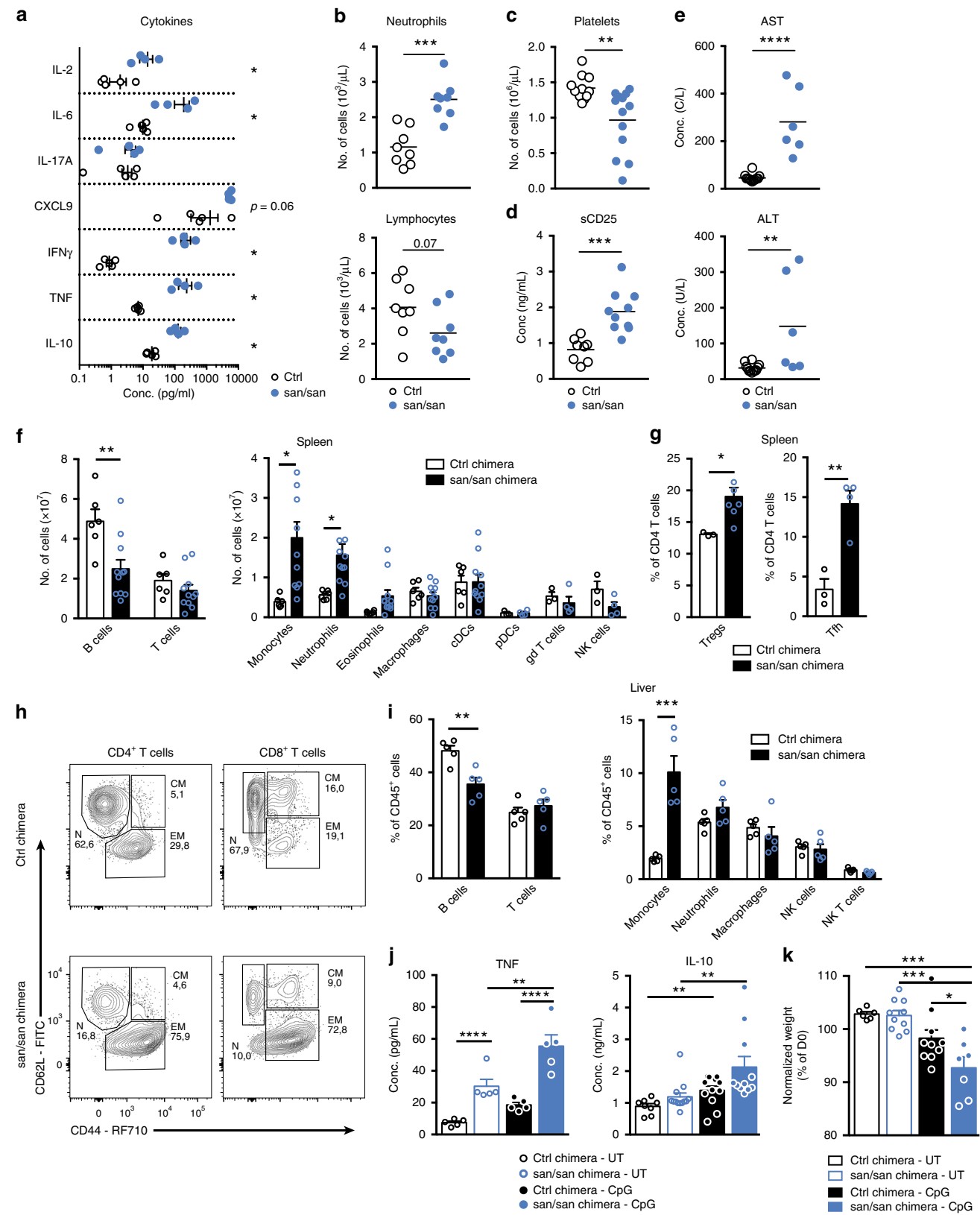

of the M199R variant (Fig. 5f, g). In contrast, sanroque monocytes and neutrophils were not increased in the chimeras (Fig. 5h, i).

**JAK1/2 inhibition reduces immunopathology in sanroque mice**. Ruxolitinib is a JAK1/2 inhibitor that is approved for the

treatment of myelofibrosis and polycythemia vera in JAK2 gain of function mutations. It inhibits a number of cytokines such as IL-1, IL-6, IL-18, IFNγ, and TNF and reduces pathology in models of HLH[39,40]. The potential of ruxolitinib for HLH treatment has been suggested in a case study of refractory HLH[41] and is under evaluation in clinical trials (NCT03533790 and NCT02400463).

**Fig. 4** Sanroque mice recapitulate some features of the R688* variant phenotype and develop severe hyperinflammation upon challenge. **a** Serum concentrations of cytokines TNF, IFNγ, IL-17A, CXCL9, IL-10, IL-6, IL-2 in sanroque mice ($n = 4$) and control littermates ($n = 5$). $*p < 0.05$ (unpaired $t$-test). **b** Number of blood neutrophils and lymphocytes in sanroque mice ($n = 8$) and control littermates ($n = 8$). $***0.001 < p < 0.0001$ (unpaired $t$-test). **c** Number of platelets in sanroque mice ($n = 11$) and control littermates ($n = 10$) $**p < 0.01$ (unpaired $t$-test). **d** Concentration of serum soluble CD25 (sCD25) in sanroque mice ($n = 9$) and control littermates ($n = 8$). $***0.001 < p < 0.0001$ (unpaired $t$-test). **e** Serum concentration of the liver enzymes aspartate transaminase (AST) and alanine transaminase (ALT) in sanroque mice ($n = 6$) and littermate controls ($n = 12$). $****p < 0.001$ and $**p < 0.01$ (unpaired $t$-test). **f** Number of splenic immune cell subsets in sanroque chimeras ($n = 11$) and control chimeras ($n = 6$). $*p < 0.05$ and $**p < 0.01$ (unpaired $t$-test). **g** Percentage of splenic regulatory T cells (Treg) and T follicular helper cells (Tfh) in sanroque ($n = 11$) and control ($n = 6$) chimeras. $*p < 0.05$ and $**p < 0.01$ (unpaired $t$-test). **h** Contour plot of CD4+ and CD8+ T cell differentiation in sanroque and control chimeras. EM: effector memory; CM: central memory; N: naive. **i** Immunophenotyping of liver derived CD45+ cells in sanroque ($n = 5$) and control chimeric mice ($n = 5$). $*p < 0.05$ and $**p < 0.01$ (unpaired $t$-test). **j** Serum concentrations of TNF and IL-10 and **k** body weight of sanroque and control chimeras treated with 50 μg ODN-1826 CpG or vehicle control every 2 days for 8 days. $*p < 0.05$, $**p < 0.01$, $***0.001 < p < 0.0001$, $****p < 0.001$ (one-way ANOVA with Dunnett's multiple comparisons test). Data shown are accumulated from three independent experiments (**d**), two experiments (**a–c**, **e**, **f**, **j**, **k**), or representative for two experiments (**g–i**). When applicable, mean and/or SEM are depicted. Source data are provided as a Source Data file

To test the role of dysregulated cytokine release in the setting of impaired Roquin-1 function, sanroque mice were treated with ruxolitinib. After 5 days of treatment, normalization of spleen size was observed with reduction of monocyte and eosinophil numbers (Fig. 5j, k). TNF and the IFNγ inducible chemokine CXCL9 similarly decreased alongside CD64 expression, a known IFNγ response gene, on monocytes (Fig. 5l). Ruxolitinib did not repress EM T cells in the spleen nor did it reduce IFNγ (Fig. 5l and Supplementary Fig. 5B). In conclusion, the results of BM chimeras and JAK1/2 inhibition demonstrate that whereas Roquin-1 directly controls T cells, splenomegaly, monocyte and granulocyte expansion are indirect consequences of dysregulated cytokine release.

**R688* mutation results in impaired localization in P-bodies.** Cytoplasmic granules such as processing bodies (P-bodies) and stress granules (SGs) are major integration sites for the regulation of mRNA fate[42]. Whereas SGs contain stalled polysomes, P-bodies are enriched in proteins that mediate RNA degradation, surveillance and translational repression[43]. As Roquin-1 is enriched within both cytoplasmic granules and its activity is correlated with P-body colocalization[8,9], we speculated that the R688* variant results in aberrant Roquin-1 localization. HEK293T cells were transfected with WT or R688* Roquin-1 and stained with antibodies to visualize P-bodies and Roquin-1. Whereas WT Roquin-1 had a speckled appearance and colocalized with Edc4+ P-bodies, distribution of the R688* mutant was more diffuse and impaired in its localization to P-bodies (Fig. 6a). Colocalization was quantified and revealed a decrease of the Pearson correlation coefficient (PCC) and Manders colocalization coefficient 1(CMM1) upon R688* Roquin-1 transfection (Fig. 6b). These results were confirmed in HEK293T and murine T cells, using DCP1 and Rck, alternative markers of P-bodies (Supplementary Fig. 6A, B). To test dominant negative behavior, WT Roquin-1 fused to GFP and V5-Roquin-1 or V5-R688* Roquin-1 were cotransfected into HEK293T cells. Similar to Fig. 6a, V5 fused WT Roquin-1 colocalized with Edc4+ granules whereas V5-R688* displayed a more diffuse appearance. The cotransfected Roquin-1-GFP retained a speckled organization that coincided with Edc4 independent of WT or R688* Roquin-1 (Fig. 6c). Quantification of colocalization confirmed that R688* mutant did not impact WT protein localization or vice versa (Fig. 6d, e). Roquin-1 accumulation in SG upon arsenite treatment was similar for WT and R688* Roquin-1 (Supplementary Fig. 6C, D), confirming previous reports that SG recruitment requires the aminoterminus of Roquin-1, harboring an intact ROQ domain[8].

**Reduced association of R688* Roquin-1 with CCR4-NOT complex.** As Roquin-1 lacks nuclease activity, it induces mRNA decay by recruiting proteins from both the decapping or deadenylation complexes through amino- or carboxy-terminal regions, respectively[9,17]. We overexpressed V5 tagged WT and R688* mutant Roquin-1 in HEK293T cells and coimmunoprecipitated Roquin-1-associated proteins with an anti-V5 monoclonal antibody. R688* Roquin-1 readily interacted with Edc4 (Fig. 6f). In contrast, association with CNOT1, the scaffold protein of the CCR4-NOT deadenylase complex, was reduced (Fig. 6f). The detection of the faint CNOT1 band (compared with control IgG), might suggest a weaker secondary binding site for CNOT1 upstream of R688 or be a consequence of a macromolecular complex comprising both the decapping and deadenylation machinery (Edc4-Rck-CNOT1 complex) bridged by Rck[44].

**ICOS mRNA decay is impaired in the presence of R688* Roquin-1.** Our results predict that deletion of the C-terminal part in R688* Roquin-1 results in a loss of post-transcriptional control. Chase experiments with actinomycin D demonstrated that the stability of *ICOS* mRNA was enhanced in R688*/R688* T cells (Fig. 6g). To address whether the reduced interaction of R688* Roquin-1 with CNOT1 impaired mRNA deadenylation, we assessed the poly(A) tail of *Icos* mRNA in 4-OHT treated *Rc3h1/2*fl/fl; CD4-CreERT2; rtTA CD4+ T cells transduced with retroviral vectors encoding doxycycline inducible WT or R687* *Rc3h1* variant (murine equivalent of R688*). Absence of Roquin-1 and Roquin-2 resulted in strongly enhanced levels of poly(A) tailed *Icos* mRNA in murine T cells (Fig. 6h). This was partially restored in cells re-expressing WT Roquin-1. In contrast, R687*-Roquin-1 failed to reduce poly(A) tailed *Icos* mRNA (Fig. 6h). These results indicate that the loss of interaction between R688* Roquin-1 and the CCR4-NOT deadenylase complex results in enhanced *ICOS* mRNA stability.

**Roquin-1 mutant comparison reveals variant specific defects.** To compare the effects of Roquin-1 variants in more detail, 4-OHT treated *Rc3h1-2*fl/fl; CD4-CreERT2; rtTA CD4+ T cells were transduced with inducible constructs encoding GFP, GFP fused WT *Rc3h1*, GFP-M199R *Rc3h1*, GFP-R687* *Rc3h1* or with GFP-*Rc3h1* 1-509AA, an aminoterminal construct representing a MALT1 cleaved Roquin-1 (Supplementary Fig. 7A, B and ref. [9]). Upon Roquin-1 and Roquin-2 deletion, ICOS expression increased dramatically (Fig. 7a). Whereas reconstituting the double-deficient T cells with GFP-WT Roquin-1 was sufficient to normalize ICOS expression, ICOS levels were not completely rectified upon introduction of the M199R, R687*, or 1-509AA Roquin-1 mutants (Fig. 7a). Quantification of ICOS fluorescence revealed that R687* and M199R reduced ICOS expression to a

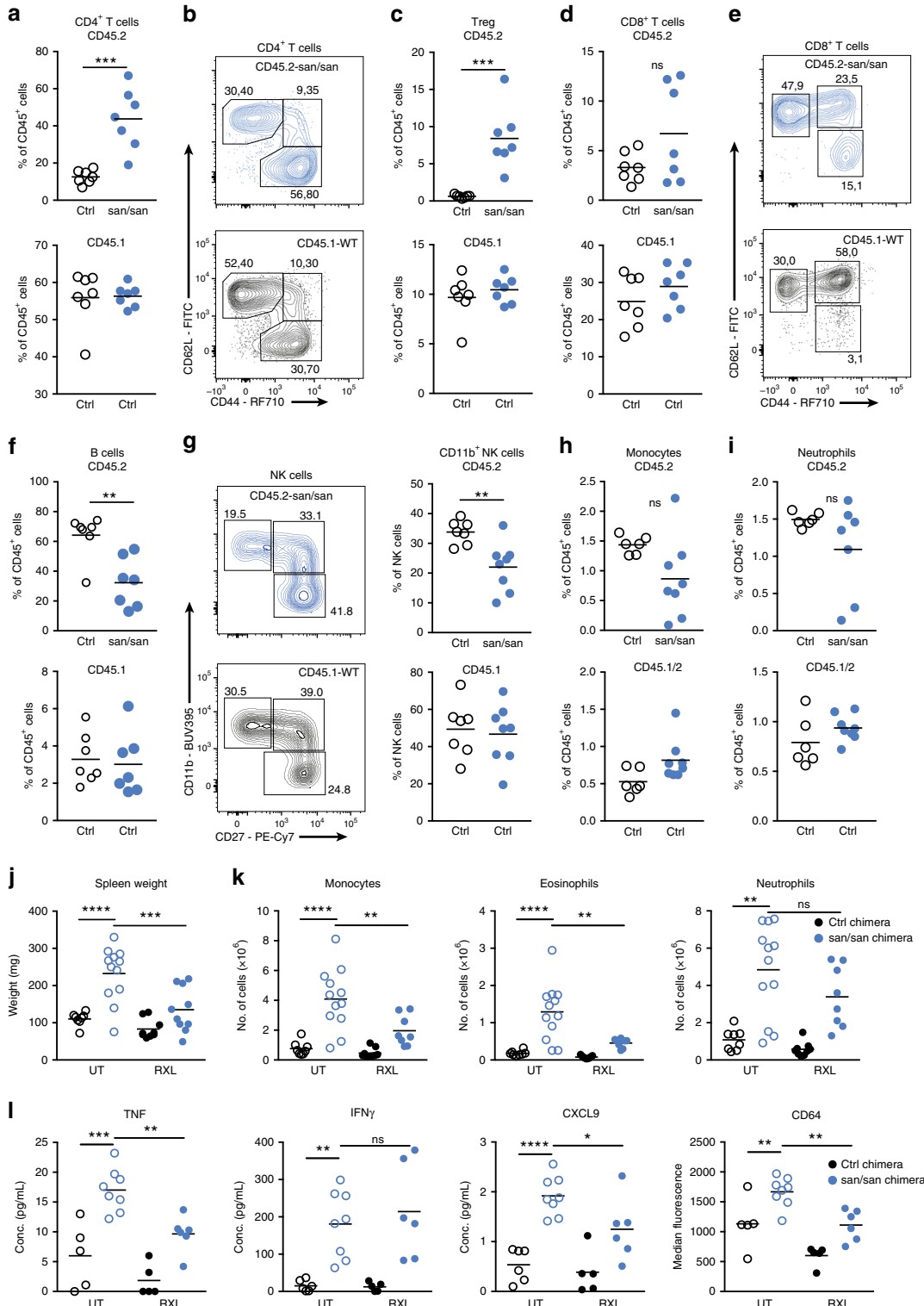

similar extent whereas the 1-509AA variant was more impaired (Fig. 7b). As doxycycline treatment resulted in supraphysiologic levels (Supplementary Fig. 7A, B), we correlated ICOS fluorescence with GFP in GFP$^{dim}$ T cells for the different Roquin-1 constructs. Fitting of regression curves generated dose response curves for each Roquin-1 variant (Fig. 7c). This revealed a stronger reduction of ICOS in cells expressing low levels of WT or M199R Roquin-1 compared with cells that express comparable

levels of the R687* or 1-509AA variants (Fig. 7c). Expression of Ox40 and CTLA4 were not repressed by the R687* and 1-509AA variants whereas the M199R mutation reduced both surface proteins to a similar extent as WT Roquin-1 (Fig. 7d, e). Similar observations were made for c-Rel (Supplementary Fig 7C). These data indicate that the M199R and R687* variants represent hypomorphic mutations but have diverging effects on specific targets.

**Fig. 5** Sanroque BM chimeras reveal cytokine driven immune dysregulation blocked by chemical JAK1/2 inhibition. **a** Percentage of CD4$^+$ T cells in mixed CD45.2$^{control}$/CD45.1 ($n = 7$) and CD45.2$^{sanroque}$/CD45.1 bone marrow chimeric mice ($n = 7$). ***$0.001 < p < 0.0001$ (unpaired $t$-test). **b** Contour plot of CD4$^+$ T cell differentiation in mixed bone marrow chimeras. EM: effector memory; CM: central memory; N: naive. **c** Percentage of regulatory T cells (Treg) in mixed CD45.2$^{control}$/CD45.1 ($n = 7$) and CD45.2$^{sanroque}$/CD45.1 chimeras ($n = 7$). ***$0.001 < p < 0.0001$ (unpaired $t$-test). **d** Percentage of CD8$^+$ T cells in mixed CD45.2$^{control}$/CD45.1 ($n = 7$) and CD45.2$^{sanroque}$/CD45.1 chimeric mice ($n = 7$). ***$0.001 < p < 0.0001$ (unpaired $t$-test). **e** Contour plot of CD8$^+$ T cell differentiation in mixed bone marrow chimeras. **f** Percentage of B cells in mixed bone marrow chimeras ($n = 7$). **$p < 0.01$ (unpaired $t$-test). **g** Contour plot of NK-cell maturation. Scatter dot plot of CD11b$^+$ NK cells in mixed CD45.2$^{control}$/CD45.1 ($n = 7$) and CD45.2$^{sanroque}$/CD45.1 bone marrow chimeric mice ($n = 7$). **$p < 0.01$ (unpaired $t$-test). **h, i** Percentage of monocytes and neutrophils in mixed CD45.2$^{control}$/CD45.1 ($n = 7$) and CD45.2$^{sanroque}$/CD45.1 bone marrow chimeric mice ($n = 7$). **j** Spleen weight in control and sanroque bone marrow chimeric mice treated with ruxolitinib (RXL) or vehicle. $n^{Ctrl\ chimera\ vehicle} = 3$; $n^{sanroque\ chimera\ vehicle} = 4$; $n^{Ctrl\ chimera\ RXL} = 3$; $n^{sanroque\ chimera\ RXL} = 3$. **$p < 0.01$ (unpaired $t$-test). **k** Number of splenic monocytes, neutrophils and eosinophils in control and sanroque bone marrow chimeric mice treated with ruxolitinib (RXL) or vehicle. $n^{Ctrl\ chimera\ vehicle} = 3$; $n^{sanroque\ chimera\ vehicle} = 4$; $n^{Ctrl\ chimera\ RXL} = 3$; $n^{sanroque\ chimera\ RXL} = 3$. **$p < 0.01$; ****$p < 0.0001$ (one-way ANOVA with Dunnett's multiple comparisons test). **l** Serum concentration of TNF, IFNγ and CXCL9; median expression of CD64 on monocytes. $n^{Ctrl\ chimera\ vehicle} = 3$; $n^{sanroque\ chimera\ vehicle} = 4$; $n^{Ctrl\ chimera\ RXL} = 3$; $n^{sanroque\ chimera\ RXL} = 3$. *$p < 0.05$; **$p < 0.01$; ***$0.001 < p < 0.0001$; ****$p < 0.0001$ (one-way ANOVA with Dunnett's multiple comparisons test). Data shown are representative of two independent experiments (**a–i**), accumulated from two independent experiments (**j**, **k**) or one experiment (**l**). When applicable, mean and/or SEM are depicted. Source data are provided as a Source Data file

**Dysregulated post-transcriptional control of cytokines.** Similarly, intracellular TNF levels were measured upon stimulation of transduced T cells in the presence of PMA/ionomycin and brefeldin A (Fig. 7f). This revealed that whereas complementation with WT Roquin-1 and M199R Roquin-1 effectively inhibited TNF production, both the 1-509AA and R687* variant failed to control TNF (Fig. 7f, g). Modeling the regulatory capacity of these variants revealed a similar activity for both the R687* and 1-509AA variant (Fig. 7h). Similarly, Roquin-1 1-509AA and R687* Roquin-1 also failed to suppress IL-2 and IL-17A upon T cell activation (Fig. 7i, j). In conclusion, our data reveal that the R687* but not the M199R variant failed to regulate the production of inflammatory cytokines.

## Discussion

In this report, we describe the consequences of a homozygous nonsense R688* *RC3H1* mutation in a patient suffering from an immune dysregulation syndrome with uncontrolled systemic inflammation. PBMC analysis reveals an increase in effector CD8$^+$ T cells, Th17 cells and Tregs, upregulation of ICOS and OX40 and a profound maturation defect in the B cell lineage. A wide range of cytokines is markedly increased regardless of Cyclosporin A (CSA) treatment. The R688* mutation of *RC3H1* lies within the proline-rich domain and produces a truncated Roquin-1 that fails to colocalize with P-bodies and is impaired to interact with CNOT1. This results in *ICOS* mRNA stabilization, increased expression of ICOS, OX40, and CTLA4 and dysregulated cytokine production. Although this study relies on a single family and one should be careful when inferring evidence from overexpression systems, cell lines and even murine models, the accumulated evidence strongly suggests a causal relationship between the R688* *RC3H1* variant and the observed disease.

The significance of Roquin-1 as a post-transcriptional regulator of immune responses is well characterized thanks to the study of a number of mouse models (summarized in Supplementary Table 2 and Supplementary Fig. 8). Comparing the R688* variant with these mouse models allows us to formulate a number of interpretations. Bertossi et al. reported that complete loss of Roquin-1 resulted in postnatal lethality and this survival deficit was later confirmed in a subsequent study[26,45]. This might suggest that the R688* variant retains some critical functions. Indeed, our data reveal that ICOS expression is still partially regulated. Analysis of Roquin mouse models suggests a role of Roquin-1 in body growth[26]. In line with these observations, the patient is short of stature, although there was no evidence of neural tube closure defects. The observed immunophenotype is reminiscent of the sanroque mice and other Roquin-1 mouse lines (Supplementary

Table 2 and Supplementary Fig. 8). We also identified some notable differences. The marked B cell maturation defect in the presence of the R688* variant has not been observed in any of the Roquin-1 transgenic mouse lines, although we observe a reduced number of B cells in the sanroque chimeras[18,23,26,45,46]. This maturation defect might be a consequence of the chronic calcineurin inhibition by CSA treatment to suppress T cell hyperactivation[47]. The number of the circulating T follicular helper cells is not increased nor does the patient present with overt signs of autoimmunity[15,23–25]. Finally, we have found increased production of IL-17A, a feature that is only observed in mice upon combined ablation of Roquin-1 and its paralog Roquin-2[19]. Although reservations should be made when comparing pathogen-free mice and immune deficient humans, we speculate that whereas the expansion of Tfh cells and GC B cells is under the control of the redundant functions shared with Roquin-2, Th17 differentiation and IL-17A production can also be dysregulated in humans in the presence of Roquin-2.

Comparing the R687* variant (the murine equivalent of R688*) and M199R mutation, we found remarkable differences in the post-transcriptional functionality (Fig. 7). This residual function of M199R variant might be crucial for autoimmune disease development. In line with this reasoning, it is of interest to note that the heterozygous parents acquired autoimmunity (Fig. 1a). Similarly, a heterozygous deletion of the last 16 exons of *RC3H1* has been detected in a Japanese patient with autoimmune disease-like symptoms with high titers of rheumatoid factors[48]. These observations strengthen the hypothesis that residual function of Roquin-1 may be required for autoimmune antibody generation. This concept merits further attention and experiments are underway to test this hypothesis.

The precise mode of action of the M199R mutation in Roquin-1 is still unsolved, but it is believed to impair an interaction with an unrecognized binding partner of Roquin-1 besides Edc4, CNOT1, or Nufip2[17,20,49,50]. Our work has now established that the carboxy-terminal truncation of Roquin-1 beyond R688 deletes sequences required for CNOT1 interaction but retains sequences with the ability to interact with Edc4. Surprisingly, targets of Roquin-1 respond to a different extent to this partial loss of posttranscriptional activity. Considering that Roquin-1 can trigger deadenylation, decapping and translational inhibition in a redundant manner[16], we propose that Roquin-1 interacts with different post-transcriptional effectors through independent modules in its polypeptide sequence. Therefore, despite the fact that a complete loss-of-function mutation may cause postnatal lethality and may therefore not be found in human patients, additional mutations similar to the one described here may exist,

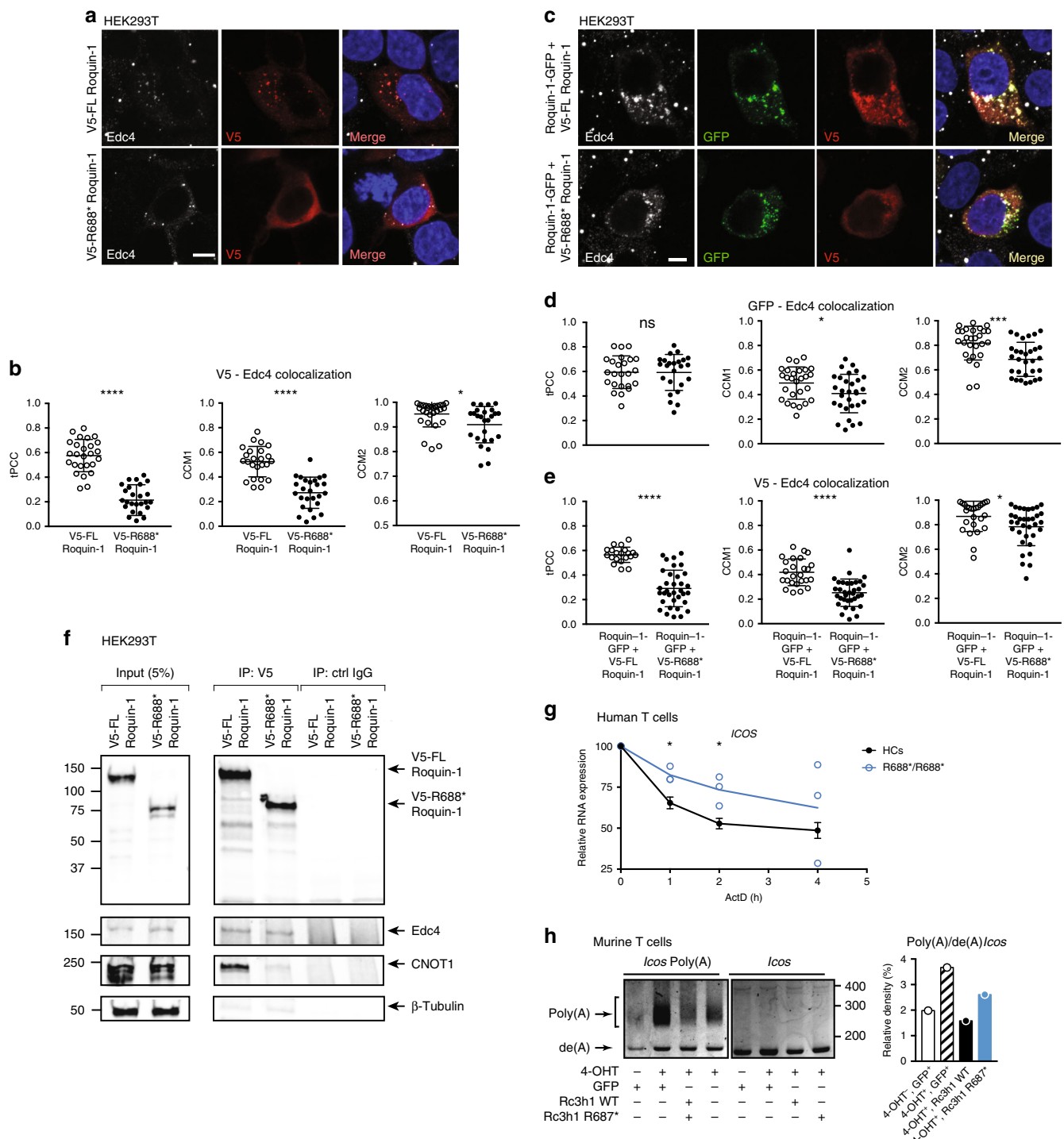

partially crippling Roquin-1 function by interfering with individual modes of post-transcriptional regulation and resulting in immune deficiencies with variable clinical phenotypes.

Sanroque mice and several Roquin-1 deficient mouse lines develop oligoclonal lymphoproliferation with effector memory T cell accumulation, resembling the immunophenotype of the R688* variant (Supplementary Table 2). Murakawa et al. used PAR-CLIP and identified Roquin-1 mRNA targets including those coding for proteins that are involved in DNA repair, cell cycle control and p53 signaling[14]. Given that these pathways have important tumor suppressor functions, additional studies are needed to assess the risk of lymphoma development particularly

in the light of findings of increased incidence of angioimmunoblastic T cell lymphomas in heterozygous sanroque mice by Ellyard et al.[51].

Immune dysregulation syndromes with hyperinflammation are the consequence of uncontrolled activation of the immune system. In the setting of familial hemophagocytic lymphohistiocytosis, the pathogenesis is dictated by perpetual immune cell activation in absence of cell mediated cytotoxicity. X-linked lymphoproliferative disorders represent a group of immune dysregulatory diseases defined by a failure to control Epstein-Barr virus infections and ensuing development of hyperinflammation. Here, we present an example of an immune dysregulation

**Fig. 6** Impaired P-body colocalization, CNOT1 interaction and *ICOS* mRNA degradation in the presence of Roquin-1 R688*. **a** HEK293T cells were transiently transfected with V5 tagged Roquin-1 (V5-FLRoquin-1) or V5-R688* Roquin-1 and subsequently stained with anti-V5 and anti-Edc4 (P-body marker). Nuclei were revealed using Hoechst. Scale bar = 10 μM. **b** Correlation analysis of Edc4 and Roquin-1 comparing V5-FLRoquin-1 ($n = 26$ cells) and V5-R688* Roquin-1 ($n = 26$ cells) transfected HEK293T cells. tPCC: tresholded Pearson correlation coefficient; CCM1/2: Manders coefficient1/2. Mean and standard deviation are plotted. $*p < 0.05$ and $****p < 0.0001$ (unpaired *t*-test). **c** HEK293T cells were transiently transfected with a combination of wild-type Roquin-1 fused with GFP and V5 tagged wild type or R688* mutant Roquin-1. Slides were subsequently stained with anti-V5, anti-Edc4 and Hoechst. Scale bar = 10 μM. **d**, **e** Analysis of correlation between Edc4 and GFP fused Roquin-1 (**d**) and V5 tagged Roquin-1 (**e**), respectively. Analysis is based on at least 24 cells/group and mean and standard deviations are plotted. $*p < 0.05$; $***0.001 < p < 0.0001$; $****p < 0.0001$ (unpaired *t*-test). **f** HEK293T cells were transiently transfected with V5 tagged Roquin-1 (V5-FLRoquin-1) or V5-R688* Roquin-1. After immunoprecipitation with anti-V5 or control IgG coupled to Dynabeads Protein G, endogenous Edc4 and CNOT1 and overexpressed Roquin-1 variants were revealed by immunoblot analysis. β-Tubulin serves as a loading control. **g** *ICOS* mRNA transcripts upon actinomycin D treatment at given time points in R688*/R688* T cells. Data was normalized using the housekeeping genes *HPRT* and *GAPDH*. HCs ($n = 11$), R688*/R688* ($n = 3$). $*p < 0.05$ (unpaired *t*-test). Mean and SEM are depicted. **h** Poly(A) tail length measured for *Icos* mRNA in murine *Rc3h1-2*^fl/fl; CD4-CreERT2; rtTA CD4+ T cells retrovirally transduced with GFP, GFP fused WT Roquin-1 or R687* Roquin-1. Bar graph represents ratios of Poly(A) tailed *Icos* mRNA over de(A) *Icos* mRNA. Data shown are representative of 2 (**f**, **h**), 3 (**a**–**e**) or accumulated data of three experiments (**g**). Source data are provided as a Source Data file

syndrome caused by the impaired function of the post-transcriptional repressor Roquin-1. The inability of the R688* variant to extinguish immune activation results in lymphoproliferation and uncontrolled cytokine release. Roquin-1 is a member of a larger family of post-transcriptional regulators of the immune system. Identification of additional variants in this family of proteins in immune dysregulation syndromes will lead to novel insights in the regulation of the immune system and define new therapeutic opportunities.

## Methods

**Human subjects**. The patient and family members provided written informed consent for participation in the study, in accordance with the 1975 Helsinki Declaration. The patient and family members formally agreed for the publication of the findings of this study. The research protocol was approved by the ethical committee of Ghent University Hospital (2012/593).

**Genetic investigations**. Karyotype analysis was performed on the index patient. Chromosomes from cultivated peripheral blood lymphocytes were analyzed with the conventional G-banding technique. Microarray-based comparative genomic hybridization (array-CGH) was performed using an 180 K oligonucleotide array with an average genome-wide resolution of ~100 kb (SurePrint G3 Human CGH Microarray Kit, Agilent Technologies). Hybridizations were performed according to manufacturer's instructions with minor modifications. Results were analyzed using Vivar. For whole exome sequencing (WES), genomic DNA was isolated from whole blood leukocytes using the Puregene DNA isolation kit (Qiagen) according to manufacturer's protocol. Whole exome enrichment was performed with the SureSelect Human All Exon V4 kit (Agilent Technologies). Paired-end massively parallel sequencing (100 cycles) was performed on a Hiseq2000 sequencer (Illumina). Data analysis was performed with our in-house developed analysis pipeline Seqplorer. In brief, read mapping against the human genome reference sequence (NCBI, GRCh37), and post-mapping duplicate read removal, quality-based variant calling and coverage analysis were performed with BCBio. Variants were annotated with Ensemble's Variant Effect Predictor (VEP) and Gemini. Variants were filtered on impact and minor allele frequency with dbNSFP. Primers for amplification and sequencing of exon 12 of Roquin-1 were designed with Primer3Plus [http://www.bioinformatics.nl/cgi-bin/primer3plus/primer3plus.cgi/]. For assessing the inheritance mode of the Roquin-1 mutation, DNA from both parents was tested.

**Mice**. Sanroque mice carrying the M199R mutation in the ROQ domain of Roquin-1[23] and backcrossed to the C57Bl/6 background were housed under specific pathogen-free (SPF) conditions. To generate bone marrow chimeras in C57Bl/6 CD45.1/CD45.2 hosts, the bone marrow (BM) of wild-type CD45.1, CD45.2 sanroque and CD45.2 control mice were used. In short, 12 h after lethal irradiation (8 Gray), CD45.1/2 mice were injected intravenously with $2 \times 10^6$ bone marrow cells. According to the experiment, BM cells of CD45.2 sanroque and control C57Bl/6 mice or mixtures of these cells together with BM cells carrying the CD45.1 analog were used. Mice were analyzed at least 8 weeks after irradiation. *Rc3h1-2*^fl/fl; CD4-CreERT2 mice[18] were crossed with rtTA transgenic mice (Jackson Laboratory) to generate *Rc3h1-2*^fl/fl; CD4-CreERT2; rtTA mice and were housed in a SPF facility in accordance with the Ludwig-Maximilians-Universität München institutional, state and federal guidelines. All experimental procedures involving mice were performed in accordance with the ethical regulations for animal testing and research of and were approved by the local government (Animal

Experimentation Ethics Committee of the Australian National University, Ethical Committee of the Ghent University Faculty of Sciences).

**Cells and media**. Peripheral blood mononuclear cells (PBMC) were isolated using Leucosep tubes (Greiner Bio) containing Ficoll density gradient medium. Cells were stored in RPMI-1640 medium, GlutaMax supplemented (Gibco, 61870044) enriched with 10% Fetal Calf Serum (FCS; Sigma Aldrich; F7524) containing 10% dimethyl sulfoxide (DMSO; Sigma Aldrich; D2650) at −150 °C, until further use. PBMC were thawed in 37 °C preheated complete medium (RPMI-1640 medium supplemented with GlutaMAX, 10% FCS, 1% penicillin-streptomycin (Pen/Strep; 10,000 U/mL; Gibco; 15140122), 1 mM sodium pyruvate (Gibco; 11360070), 1% non-essential amino acids (NEAA; Gibco; 11140035) and 50 μM 2-mercaptoethanol (Gibco; 31350010). In the setting of functional testing, cells were left to recuperate for 30 min at 37 °C and 5% $CO_2$ after removal of DMSO. CD4+ T cells, CD8+ T cells and CD14+ monocytes were enriched using positive selection of abovementioned populations with microbeads and magnetic assisted cell separation according to manufacturer's protocols (Miltenyi; 130-045-101 (CD4), 130-045-201 (CD8), or 130-050-201 (CD14)). PHA blasts were generated from PBMCs using 1% PHA (v/v; Life Technologies; 10576015) and 20 ng/mL IL-2 (eBioscience; PHC0021). The cells were cultured at a density of $2 \times 10^6$/mL in complete medium at 37 °C and 5% $CO_2$ for 4–14 days. Every 2–3 days, 10 ng/mL IL-2 in fresh medium was supplemented. In the case of anti-CD3/CD28 stimulation, cells were cultured in the presence of 2 μg/mL anti-CD3 (BioLegend; 317315), anti-CD28 (BioLegend; 302914) and IL-2 in 6-well plates coated with 5 μg/mL goat anti-mouse antibody (Invitrogen; 16-5098-85).

Murine CD4+ T cells were isolated from spleens and lymph nodes of *Rc3h1-2*^fl/fl; CD4-CreERT2; rtTA mice using the EasySepTM Mouse CD4+ T cell Isolation Kit (Stem Cell; 19852) and treated with 1 μM 4′ OH-tamoxifen (4-OHT) (Sigma Aldrich, H7904) in T cell medium (DMEM (Gibco; 10566016) supplemented with 10% FCS (Gibco), 1000 U/mL Pen/Strep (Gibco), 10 mM HEPES (pH 7.4; Gibco; 15630080), 50 μM 2-Mercaptoethanol (Gibco), and 1% NEAA (Lonza) for 24 h to induce CRE enzyme mediated recombination of loxP targeted sequences at a concentration of $1 \times 10^6$ cells/mL. Afterward, CD4+ T cells were washed twice with T cell medium to remove 4-OHT. T cells were subsequently stimulated in Th1 skewing conditions by addition of 0.5 μg/mL anti-CD3 (145-2C11; in-house production Helmholtz Zentrum München), 2.5 μg/mL anti-CD28 (37.N; in-house production Helmholtz Zentrum München), 10 μg/mL anti-IL-4 (11B11, in-house production Helmholtz Zentrum München), 10 ng/mL IL-12p70 (BD Pharmingen, 554592), and subsequent culturing in 6-well plates coated with 0.05 μg/mL goat anti-hamster IgG (MP Biochemicals; 0855397) at an initial cell density of ~$4 \times 10^6$ cells/mL. After 40 h of T cell activation, T cells were transduced with retroviral particles using spinoculation (1 h, 18 °C, 850 × g). After an additional 4–6 h co-incubation of T cells and virus, viral particles were washed off and T cells were resuspended in T cell medium supplemented with IL-2 (20 U/mL). To induce construct expression, transfected T cells were cultured for 16 h in the presence of doxycycline (1 μg/mL).

HEK 293T cells were obtained from the American Type Culture Collection (ATCC; CRL-3216) and were maintained in Dulbecco's modified Eagle's medium containing 10% fetal bovine serum, 1% pen/strep at 37 °C and 5% $CO_2$.

**Plasmids and cloning**. For studies of Roquin-1, the murine equivalent R687* mutation was introduced with the Quikchange site-directed mutagenesis protocol (Agilent Technologies, Santa Clara, CA, USA) using pKMV-Roquin-1-GFP (*Roquin1-IRES-GFP*, ref. [23]) or pKMV-V5-Roquin-1 (pKMV *V5-Roquin1*, ref. [27]) as a template. For confocal imaging, HEK293T cells plated into 6-well plates were transiently transfected using 300 ng branched 25 kDA polyethylenimine (PEI) and 200 ng DNA. For immunoprecipitation experiments, subconfluent HEK293T cells were transfected 10 μg of DNA complexed with 15 μg of PEI.

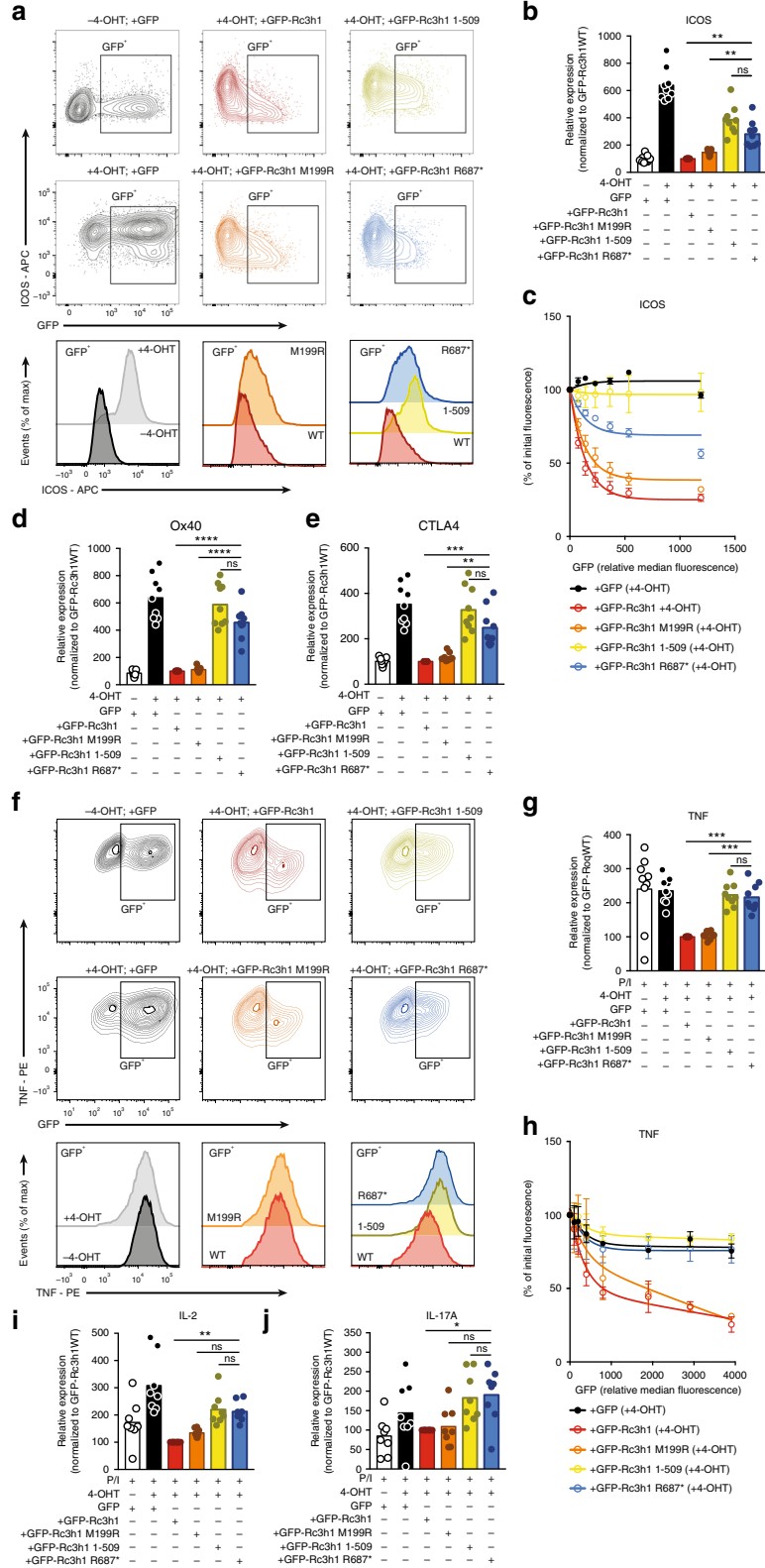

In the case of murine T cell studies, retroviral vectors (pRetro-Xtight; Clontech) expressing GFP fused Roquin-1 variants under the control of the Tet-On system were used. To create the plasmids encoding various pRetro-Xtight-GFP-Roquin-1 variants, the mutations were introduced using the Quikchange protocol. For the generation of viral particles, HEK293T cells, pretreated with 25 μM chloroquine, were cotransfected with 5 μg of pCL-Eco (Addgene; 12371) and 50 μg of the respective pRetro-Xtight-GFP-Roquin-1 variant using calcium phosphate as a transfection reagent. After 6 h of incubation with the DNA-calcium phosphate precipitates, cells were washed and cultured in fresh medium for an additional 48 h

while virus particles were collected. Viral particles were filtered (45 μM) and mixed with polybrene (10 μg/mL) prior to T cell transduction.

**Reagents**. Cells were treated with PMA (81 nM in human or 20 nM in murine studies; Sigma Aldrich; P8139) and ionomycin (1 μM; Sigma Aldrich; I3909) for indicated time points. Mepazine acetate (Vitas-M Laboratory Ltd; STK386548) was used at a concentration ranging between 6.5 and 20 μM. Sodium arsenite (Sigma Aldrich; S7400) was dissolved in complete medium at a final concentration of 1 M and used at 1 mM. Cultured monocytes were stimulated with 100 ng/mL LPS

**Fig. 7** The murine equivalent Roquin-1 R687* fails to regulate proinflammatory cytokines TNFα, IL-2 and IL-17A. **a** Contour plots showing ICOS expression in 4-OHT treated murine $Rc3h1-2^{fl/fl}$; CD4-CreERT2; rtTA CD4$^+$ T cells retrovirally transduced with GFP fused Roquin-1 variants. Representative histograms depict ICOS levels in GFP$^+$ T cells expressing various Roquin-1 variants. **b** Scatter dot plot representing mean ICOS fluorescence in GFP$^+$ T cells ($n = 9$). **c** Nonlinear regression of ICOS expression in murine T cells transfected with GFP fused Roquin-1 variants ($n = 3$). Mean and SEM are depicted. **d, e** Scatter dot plot representing mean Ox40 (**d**) and CTLA4 (**e**) fluorescence in GFP$^+$ T cells ($n = 9$). $**p < 0.01$; $***0.001 < p < 0.0001$; $****p < 0.0001$ (one-way ANOVA with Tukey's correction). **f** Contour plots showing TNF production 4-OHT treated murine $Rc3h1-2^{fl/fl}$; CD4-CreERT2; rtTA CD4$^+$ T cells retrovirally transduced with GFP fused Roquin-1 variants and treated with PMA/ionomycin for 2 h and incubated for an additional 2 h after brefeldin A supplementation. Representative histograms depict TNF expression in stimulated GFP$^+$ T cells. **g** Scatter dot plot representing mean TNF fluorescence in GFP$^+$ T cells transduced with various Roquin-1 variants ($n = 9$). **h** Nonlinear regression of TNF fluorescence in murine T cells transduced with GFP fused Roquin-1 variants ($n = 3$). Mean and SEM are depicted. **i, j** Scatter dot plot representing mean IL-2 (**h**) and IL-17A (**i**) fluorescence in GFP$^+$ T cells ($n = 8$). $*p < 0.05$; $**p < 0.01$; $***0.001 < p < 0.0001$ (one-way ANOVA with Tukey's correction). Data shown are representative for four independent experiments (**a, f**), accumulated data from two independent experiments (**c, h**) or four independent experiments (**b, d, e, g, i, j**). Source data are provided as a Source Data file

(Invivogen; tlrl-3pelps) or 5 mM ATP (Merck; A6419) dissolved in sterile H$_2$O. For the chase experiments with actinomycin D (Merck; A9415), a final concentration of 5 μg/mL was used. Fifty micrograms of CpG ODN-1826 (Invivogen; tlrl-1826) was dissolved at a concentration of 500 μg/mL in sterile H$_2$O and injected intraperitoneally every 2 days for 10 days. Ruxolitinib (ABCR; AB358151) was dissolved in 2.5% DMSO, 33% PEG400, and sterile H$_2$O at a final concentration of 6.25 mg/mL. Mice were orally gavaged twice daily during 5 days with 1.25 mg of ruxolitinib or vehicle.

**RNA isolation, qRT-PCR, and primers**. PHA blasts ($2 \times 10^6$ cells) were lysed into RLT Plus-buffer (1048449; Qiagen) and stored at −80 °C until further processing. RNA was obtained using the RNEasy Kit (74106; QIAGEN) following manufacturer's instructions. Concentration and purity of RNA was assessed using the NanoDrop 8000 technology (ThermoFisher Scientific, ND-8000-GL). Five hundred nanograms of RNA was transcribed to cDNA using the sensifast cDNA synthesis kit (Bioline; BIO – 65054) and 15 ng cDNA (estimated from input RNA) was used as input for quantitative Real-Time PCR (Lightcycler 480, Roche). Gene expression was analyzed using qbase + software version 2.6 (Biogazelle). All primer sequences can be retrieved in Supplementary Table 3.

**Poly-A tail length assay**. 4-OHT treated murine $Rc3h1-2^{fl/fl}$; CD4-CreERT2; rtTA CD4$^+$ T cells were transduced with constructs encoding GFP fused WT or R688* Roquin-1 and treated with doxycycline (1 μg/mL) for 6 h. 300 K to 1 M GFP$^+$ cells were sorted using FACS ARIA IIu. RNA was extracted using Nucleospin RNA (Macherey-Nagel; 740955) according to manufacturer's instructions. To assess poly(A) tail length, the assay was performed according to manufacturer's guidelines (ThermoFisher; 764551KT).

**Western blotting**. PHA blasts were lysed at a concentration of $2 \times 10^6$ cells in 50 μl E1A lysis buffer (1% NP40, 20 mM HEPES, pH 7.9, 250 mM NaCl, 1 mM EDTA) complemented with protease inhibitors (Complete-ULTRA; 05 892 970 001; Roche). Prior to SDS-PAGE, samples were spun at $12,000 \times g$ to remove insoluble material and were resuspended in 14 μl of loading dye. Equal amounts of protein (30 μg) were separated on a 4–15% agarose gel (Criterion TGX Stain-Free Protein Gel; Bio-Rad; 5678084) followed by semi-dry transfer to nitrocellulose. Proteins were visualized by chemiluminescence (SuperSignal West Femto; ThermoFisher; 34094). Antibodies used recognize Roquin-1/2 (Millipore; 3F12), CNOT1 (Proteintech; 14276-1-AP), Edc4 (Cell Signaling Technology; 2548), β-Tubulin (Abcam; ab21058), and β-Actin (Santa Cruz; c4, sc-47778). All uncropped images can be retrieved in the Source Data file.

**Immunoprecipitation**. Dynabeads Protein G (ThermoFisher Scientific; 10003D) were complexed with 10 μg anti-V5 (Life Technologies; 46-0705) or IgG control (BD Biosciences; 349050). HEK293T cells, transiently transfected with pKMV V5-Roquin-1 or V5-Roquin-1 R688* using PEI reagent and subsequently lysed in 350 μl E1A buffer complemented with Complete-ULTRA. 1 mg of cell lysate was incubated with anti-V5 antibody/Dynabead Protein G complexes for 1 h at 4 °C on a rotating wheel. After fixation on the magnet, the beads were thoroughly washed for 3 consecutive times with E1A buffer. The beads were resuspended in 30 μl E1A buffer and 10 μl Laemlli buffer and stored at −80 °C until immunoblotting.

**Flowcytometry**. Both PBMCs and murine cells were labeled with monoclonal antibodies labeled with fluorochromes or biotin recognizing surface markers. A complete list of the used antibodies can be found in Supplementary Table 4. In general, cells were first stained with FcR block (human; Miltenyi; 130-059-901, mouse; in-house developed; 2.4G2) together with biotin conjugated antibodies and Fixable Viability dye eFluor 506 (eBioscience; 65-0866-14), Fixable Viability Stain 620 (BD Biosciences; 564996) or blue fluorescent reactive dye (Invitrogen; L34962). In a second step, remaining surface markers were stained with a mixture of

antibodies in FACS buffer (DPBS pH 7.4, 1% Bovine serum albumin, 0.05% NaN$_3$, 1 mM EDTA). If staining of intracellular antigens was required, cells were fixed 30 min in 2% paraformaldehyde at room temperature and subsequently permeabilized with FoxP3 permeabilization buffer (eBioscience; 00-5523-00). Acquisition and analysis of labeled cell suspensions was performed with a LSR Fortessa or a BD FACSymphony flowcytometer (BD Biosciences) and subsequent analysis of data with FlowJo10 software (BD Biosciences) and R (version 3.5.1). Gating strategies of data presented in this paper can be found in Supplementary Figs. 9 and 10.

For analysis of cells via the Image Stream, cells were stained as described above and for measurement resuspended in FACS buffer containing DAPI for nuclear stain. The samples were measured with the AMNIS image stream (Millipore) and similarity score of proteins was calculated using the IDEAS software Bright Detailed similarity feature R3.

**Cytokine quantification**. Human serum cytokines IL-1α, IL-1RA, IL-6, IL-10, IL-18, IFNγ, and TNF were quantified by magnetic bead-based multiplex assay using Luminex technology (Bio-Rad) according to manufacturer's protocol with small adaptations. Serum IL-17A was measured using eBioscience Ready-Set-Go ELISA kits (ThermoFisher Scientific; 88-7371-88).

For intracellular cytokine staining, $1 \times 10^6$ PBMC were cultured in complete medium with PMA (82 nM) and ionomycin (1 μM) in the presence of brefeldin A (20 μg/mL; Sigma Aldrich; B7651) for 18 h.

For cytokine secretion, $4 \times 10^5$ PHA stimulated CD4$^+$ T cells, $4 \times 10^5$ CD8$^+$ PHA stimulated T cells or $2 \times 10^5$ monocytes were cultured 10 in complete medium without or with PMA/ionomycin (T cells) or LPS (monocytes). Supernatants were stored at −20 °C until quantification of TNF and IFNγ by ELISA.

Qualification of mouse IL-2, IL-4, IL-6, IL-10, IFN-γ, and TNF in the plasma from sanroque mice or wild-type littermate controls was performed using Meso Scale Discovery V-PLEX custom assay or the U-PLEX Biomarker assay for IL-17A and IL-21 according to the manufacturer's instructions (Meso Scale Discovery; K152A0H-2, K15069L-2). Serum levels of CXCL9 and IL-2Ralpha were quantified using the mouse SimpleStep ELISA kit (Abcam, ab203364) and mouse IL-2Ralpha DuoSet kit (R&D Systems, DY2438), respectively.

**Confocal imaging**. HEK cells were seeded in 8-well chamber slides (Ibidi). Ninety percent confluent cells were transfected using PEI[52] with indicated combinations of pKMV-Roquin-1-GFP, pKMV-V5-Roquin-1, and pKMV-V5-Roquin-1 R688* (with a fixed total of 300 ng plasmid DNA per well). Alternatively, cells were seeded onto coverslips and transfected in six-well plates with 3 μg total plasmid DNA using Lipofectamine 2000. The next day, some wells were treated with 1 mM sodium arsenite for 45 mins to induce stress granule formation. Cells were fixed with 4% PFA and cells were stained with mouse anti-V5, rabbit anti-Edc4 or anti-eIF3 (Cell Signaling Technologies; #3411) in triton (0.2%) containing staining buffer. All primary antibodies were used at a dilution of 1:100 and subsequently detected using donkey anti-mouse-AF594 or donkey anti-rabbit 650 (Molecular Probes, Invitrogen; 1:500). Untagged WT and R688* Roquin-1 were detected using an anti-Roquin-1 antibody (Novus Biologicals). Confocal images were captured with a Zeiss LSM780 confocal microscope (Zeiss, Zaventem, Belgium). Images were taken using a 63 × Pln Apo/1.4 oil objective. The pinhole was set at 1Airy Unit and scans ware made with a pixel dwell time of 2.62 μs. The scan area covered 800 by 800 pixels. Combined with a zoom of 1.9 this resulted in a pixel size of 0.089 μm. A Z-stack of 3–5 slices was recorded with a z-interval of 1 μm. Extended focus images were made in Volocity 6.3 (Perkin Elmer). Data sets for colocalization analyses were collected on an observer Z.1 microscope equipped with a yokogawa disk CSU-X1 (Zeiss, Zaventem, Belgium). Per condition z-stacks of 30 Roquin-1$^+$ cells were created with a z-interval of 0.220 μm. Parameters such as detector gain, laser intensity, exposure time, and image post-processing were kept consistent between the different conditions. A voxel-based measurement of the tresholded PCC and Manders coefficients M1 and M2 was carried out in Volocity 6.3.0 (Perkin Elmer).

**Statistics and reproducibility**. No estimate of variation has been performed within each group of data prior to statistical analysis. Data sets were analyzed using the parametric unpaired or paired *t*-test to compare two populations (when indicated). In the case of more than two populations, one-way ANOVA combined with Tukey's multiple comparisons test to correct for multiple comparisons was applied. All tests were performed as two-sided. Results with a *p* value of 0.05 or less were considered significant. Mean values, standard error of the mean and statistics were calculated with Prism7 (GraphPad software). No criteria of inclusion or exclusion of data were used in this study. Experiments were performed without prior randomization of the animals and without blinding. No statistical method was used to predetermine sample size of mouse experiments.

**Reporting summary**. Further information on research design is available in the Nature Research Reporting Summary linked to this article.

## Data availability

All sequence data pertaining to the patient and family member is stored on the secured server of the Center for Medical Genetics Ghent due to privacy regulations. This sequence data is available to researchers on request. Furthermore, all other data if not mentioned differently are contained within the article and its supplementary information or available upon reasonable request. The flowcytometry data used for FlowSOM analysis of the R688*/R688* has been deposited in the public database FlowRepository (www.flowrepository.org) as a dataset with ID: FR-FCM-Z267. The source data underlying the figures and supplementary figures of this articles are provided as a Source Data file.

## Code availability

The R-code used for the FlowSOM analysis of R688*/R688* PBMCs is publicly available and can be accessed via GitHub (https://github.com/saeyslab/Roquin-1-hyperinflammation).

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

## Acknowledgements

The authors thank the patient and his family for participating in the study and Veronique Debacker, Nancy Decabooter and Juliane Klein for excellent technical assistance. The authors are also thankful to Peter Lane and Maher Nawaf for providing bone marrow used in the sanroque mouse studies. We acknowledge the staff of the Australian Phenomics Facility, the ACRF Biomolecular Resource Facility and the Microscopy and Cytometry Facility at the Australian National University, the BMC Core Facility Flow Cytometry at the Ludwig Maximilian University for technical expertise. We would like to thank the VIB Flow Core and VIB Bioimaging Core for training, support and access to the instrument park. This work was supported by the Jeffrey Modell Foundation, the University Hospital Ghent Spearhead Initiative for Immunology Research and by grant G044615N from the Fund for Scientific Research Flanders (FWO). This project has received funding within the Grand Challenges Program of VIB. This VIB Program received support from the Flemish Government under the Management Agreement 2017–2021 (VR 2016 2312 Doc.1521/4). S.J.T. is a beneficiary of a postdoctoral BOF grant and is a postdoctoral fellow with the Fund for Scientific Research Flanders (FWO, 12W9119N). V.A. was funded by a NHMRC Centre for Research Excellence Grant (APP1079648) whilst a NHMRC Program Grant (APP1113577) and a NSFC grant (81873879) to C.G.V. funded experimental work. S.V.G. is a postdoctoral fellow with the Fund for Scientific Research Flanders (FWO, 12W9119N) and an ISAC Marylou Ingram Scholar. V.H. received DFG funding from SPP-1935, SFB-1054 TP-A03, and HE3359/4-1, as well as grants from the Fritz Thyssen and Else Kröner-Fresenius foundations.

## Author contributions

S.J.T., V.A., G.B., M.D. and F.H. designed the experiments, collected and interpreted immunological and molecular data. P.V. and B.M. discovered the mutation. P.V. and F.H. designed the study and collected and interpreted clinical data. D.B., L.N., M.D.B., E.P., J.E., J.C., H.V.G., G.V.I. and J.S. collected immunological data. S.V.G. helped with the analysis of the immunological data. M.L., R.V.C., P.S., J.D., V.B., B.N.L., C.V., V.H., R.B. and F.H. provided scientific guidance. S.J.T. and M.D. drafted the first paper. All authors revised the paper and gave their final approval of the version to be published.

## Competing interests

The authors declare no competing interests.
