## [Peer Review File · Nature Communications]

Reviewers' Comments:

Reviewer #1:

Remarks to the Author:

Review comments

Tavernier et al provide the first description of a human patient with homozygous RC3H1 loss of function variant encoding encodes Roquin-1 (p.R688*). The patient developed immune dysregulation with HLH features. The findings were compared with mouse models of encodes Roquin-1 and Roquin-2. The authors conclude that RC3H1 mutations in humans result in defective B and T cell differentiation and an inability within the innate and adaptive immune system to suppress the production of various cytokines due to the impaired interaction with an mRNA deadenylation/decapping complex. Although only based on a single patient, the article is of great interest. The flow cytometric analysis with FlowSOM is impressive whereas the molecular angle of the paper is lacking in comparison. The authors explore and discuss preclinical treatment options with ruxolitinib.

Major comments:

1. The Family history suggests a predisposition to immune pathology since the mother suffers from systemic lupus erythematosus and Sjögren's syndrome and the father from uncharacterized arthralgia and vasculitis >>> is there any indication for a haploinsufficiency with milder phenotype?
2. In Figure 2, cluster 12 is not explained even though the expression is significantly overrepresented in the patient and could be of interest. Please include not only cluster 12, but all markers and expression levels for all significant clusters in the supplementary material, or the dataset should be deposited.
3. In Figure 6C, in the condition Roquin WT GFP + Roquin R688 V5, the V5 is less fluorescent and described as "diffuse". However, in Supp. figure 5A, a speckled appearance exists. This indicates that a less strong colocalization image was used for the analysis to draw the conclusion of no colocalization. This is unsurprising given that Edc4 is IP'ed with the mutant. Please perform additional experiments to conclude that there is no colocalization of Edc4-Roquin R688*, at the moment this is not convincing.
4. In Supp. Figure 1J, mepazine treatment does not appear to reverse the degradation of Roquin-1 R688* as stated. Please provide densitometry analysis of this.
5. In Figures 7G and 7I no statistics provided due to small sample size - it requires replication. Is this consistent with Supp. Figure 6D and 6E ? please explain the discrepancy.

Additional points

1. The authors should carefully revisit the nomenclature where human and mouse genes/gene products are described.
2. 18-year-old boy ->> inappropriate to be called boy, please revise
3. Please provide an illustration of all mutations in different domains of Roquin-1 described in mouse.
4. When describing Figure 2C in the text, please specify NK cells as CD16+ or CD56+.
5. In FlowSOM analysis, it is suggested that manual gating did not identify reductions in naïve CD4 T cells as indicated by the clusters. If the number of clusters is reduced, does this issue persist?
6. The aggregation of CD20/CD14 and CD3/CD14 cells is very interesting, please describe the markers expressed on each in these aggregates, do they differ between the patient and controls?
7. In Figure 5K description, a "trend" is noted. Please remove this as it is misleading - the error bars are too large.
8. The Figure 6 description in the text is incorrect as the IP WB in 6F was clearly in 6A previously and it was not adjusted for in the text.
9. In methods, mepazine was used in the range of 6.5uM-20uM. Please show that this concentration is sufficient - MALT1 WB or flow cytometry.

10. In Figure 1C, what is the band running at about 80kDa?
11. In Figure 1, please include Sanger sequencing result of the patient and parents to show segregation of the signal.
12. In Figure 3, please include other Roquin-1 targets apart from TNF (ex IL6, IRF4). Also, why does TNF expression at the transcript level appear the same between control and patient?
13. In Figure 4J, the values for cytokines are expressed in OD. Were appropriate standard curves used? Please express this as pg/mL.
14. In Figure 4K, significance bars are duplicated, please amend.
15. In Figure 6F, the edc4 band of IgG control seems to have a smear with a band coming up right after the cut. Please do not cut that out/please show the full WB.
16. In Supp. Figure 6D-F, please provide pg/mL concentration (not normalized).

Reviewer #2:

Remarks to the Author:

Dr. Haerynk and colleagues present functional analysis of a homozygous nonsense Roquin-1 variant identified in a patient with a recurrent hyper inflammatory syndrome. This is a fascinating clinical case, and the authors provide compelling data to support a functional role for the RC3H1 variant as a driver of immune dysregulation.

The authors comprehensively characterize protein expression and localization as well as immune populations in the patient. They model the mutation with paralogous mutation in mouse with many overlapping features. Overall, this is a well-performed and well-written study.

There are a few suggestions/questions:

1. Some HLH-associated data are missing for the index patient. If there is stored plasma, are the authors able to report IL18, IL18-bp, CXCL9 levels?
2. Outcome of the patient is not described. Given the mouse response, a trial of ruxolitonib or similar agent seems reasonable. Patient response to such an intervention would greatly increase the scientific and clinical impact of this study. Did this patient undergo stem cell transplant?
3. In Figure 5, response of immune cell populations to JAK1/2 inhibition is described. It would be informative to also include cytokine responses.

Reviewer #3:

Remarks to the Author:

Tavernier et al describe an interesting observation based on a single patient carrying a homozygous point mutation (R688*) of human Roquin that results in the expression of a C-terminally shortened form of Roquin. This patient suffers from periodic hyperinflammation and the authors have performed multiple experiments to link this clinical phenotype to the mutant Roquin protein. There is extensive knowledge defining Roquin as a (negative) posttranscriptional regulator of inflammatory mRNAs. Hence, the study was designed to (i) phenotype the immunopathological syndrome of the patient at high resolution by assessing immune cell subpopulations and by determining changes within the cytokine milieu ex vivo and in vitro, (ii) relate this to the phenotype of an already described mutant Roquin mouse strain carrying another point mutation (M199R), and (iii) establish a mechanism for the R688* Roquin mutant by assessing its subcellular localization and interaction with EDC4 and CCR4. Despite these efforts, the data sets remain descriptive and mostly do not reveal direct effects or a mechanism. Key experiments such as reverting the mutation in patient-derived cells by genomic engineering to proof that increased cytokine levels become normalized are missing. Reconstitution of murine Roquin-deficient T-cells

with R688* variants fails to show a function of this mutant for TNFa, IL-17a and IL-2 expression. No experiments were performed to prove that R688* affects mRNA deadenylation or mRNA decay in model systems (such as HEK293 cells used in Fig. 6) or even in immune cells. A confounding problem is that in many experiments multiple healthy controls were evaluated against a single sample only which was available from the patient. Furthermore, comparing the data presented in Fig.2 versus Fig. 3A, an alternative explanation for the patient's clinical symptoms would be that there are defects in the myeloid system that result in periodic release of all major proinflammatory cytokines to the systemic circulation and this then causes long-term secondary differentiation defects in immune cell subpopulations.

Specific comments:

Fig.S1J: the protein band intensities of full length Roquin-1, R688* Roquin-1 and the cleaved Roquin fragment need to be quantified from replicate immunoblot experiments. As shown, the proposed increase in cleaved Roquin forms (including R688*) upon PMA/Ionomycin treatment and the inhibitory effects of Mepazine are not particularly obvious and seem to be inconsistent between conditions.

Fig.2, Fig. S2. These two figures are hard to follow for non-immunologists not familiar with all the different immune cell populations that can be defined by sets of surface markers. Multiple subsets of immune cells identified by FACS analysis are discussed in the text but are not shown in the figures (e.g. clusters 194, 145, 91). The data reveal very complex shifts in the T-, B-, monocyte and NK cell repertoire. However, no functional analysis is provided to demonstrate that these immune cells have indeed altered cytokine profiles or cytotoxic activities or other loss- or gain-of-function phenotypes. These data need to be restructured and focused on the strongest differences between healthy controls and the patient. The basis of the in silico approach to cluster flow cytometry data needs to be explained for non-specialists. By which criteria were some sorted populations manually annotated? How can it be ruled out that the changes have developed slowly over time (or due to immunosuppressive treatment by CSA) and are thus not directly related to the R688* mutation?

Fig. 3A: There are profound increases in serum levels of IL-1b but also the IL-1R antagonist (IL-1RA). Production of IL-1b alone would be sufficient to cause a hyperinflammatory syndrome, but this possibility is not discussed at all. Also how an altered balance of IL-1beta and IL-1RA could contribute to the disease phenotype is not discussed. Similarly, arguments apply for elevated serum levels of IL-18. In the light of these observations, the reason why the authors focus largely on adaptive immune cells remains unclear.

Fig. 3B: related to this point: activation of T-cells in vitro by artificial addition of PHA/IL-1 only doubles expression of TNFa/IFNg, whereas in the blood samples changes of cytokines mainly released from the myeloid compartment increase at the log scale!

Fig. 3E: this figure shows inducible mRNA steady state levels of TNFa in both, the healthy controls and the patient samples. The suppressive effect of Mepazine on controls cannot be interpreted as altered mRNA stability as discussed in the text unless results from mRNA decay assays are provided. The mRNA steady state levels for TNF and IFNg mainly show comparable activation in the patient samples and weak (if at all) or variable effects after Mepazine treatment.

Fig. 4/5: The necessity to include the extensive data set from the sanroque (M119R) mice can be questioned. Multiple parameters do not really match the patient's characteristics and the authors eventually conclude that the reduced Roquin function in these mice results in a more pronounced MAS-like phenotype; hence altogether these mice simply have a different immune pathology. Along the same lines: the bone-marrow transplantation and JAK1/2 inhibitor experiments are interesting but do not help to unravel the R688* phenotype.

Fig. 6: At this point of the story, a turn is made to reveal a mechanism for R688* Roquin. Fig. 6A is not mentioned in the text. Fig. 6A-E show altered distribution of Roquin to EDC4-positive granules. No other markers were included to prove that these are indeed P-bodies. P-bodies are now largely viewed as storage sites for mRNAs stalled in translation rather than as sites of specific mRNA decay (see Hubstenberger, A., et al., P-Body Purification Reveals the Condensation of Repressed mRNA Regulons. Mol Cell, 2017. 68(1): p. 144-157). Therefore, the altered P-body localization of R688* Roquin is not indicative of altered mRNA decay functions. Why were these experiments only performed in HEK293 cells and with overexpressed proteins?

Fig. 6F: The observation of loss of the CNOT1 subunit from R688* Roquin immune complexes is important. Does this R688* Roquin complex has less deadenylase or decapping activity? Clarifying this point is crucial and several assays are available to determine polyA tail length of model transcripts or endogenous mRNAs.

Fig. 7/S6: The reconstitution experiments of conditional Roquin1/2-deficient murine T-cells were well performed and controlled. These data are convincing, but in the end they do not uncover a role of R687* (the murine equivalent of R688*) for the regulation of major cytokines (TNF α , IL17A and IL-2) or OX40/ICOS implicated in the patient's disease (see Fig. 2F-K).

Reviewer #4:

Remarks to the Author:

Tavernier et al.

Nature Communications manuscript NCOMMS-19-03035-T

In this paper, the authors describe the first human case (according to the authors) with a homozygous mutation in RC3H1, encoding Roquin-1, a well-known modulator of cytokine mRNA stability. They identified this mutation by performing whole-exome sequencing of genomic DNA from this subject, who presented with an HLH-like syndrome, but who did not have evidence of other mutations in known HLH genes. They have performed a detailed evaluation of this subject, and compared the clinical and molecular phenotypes to those seen in mice with various types of induced mutations in the orthologous gene.

This is a very interesting, complete and well-written paper, and I think it is deserving of publication in a general and high-profile journal such as Nature Communications. There are a few aspects that I think would improve the quality of the paper and make it more accessible for a general audience.

General points:

1. Page 4, first paragraph. The word "symptoms" should be reserved for aspects of the disease that are subjectively felt by the patient, as opposed to signs.
2. P. 5, second paragraph. If the authors are going to use the Gene-/- terminology, they should use the correct gene name – if they are going to use the protein names, they should figure out another method of identifying the mutant or normal proteins.
3. Figure 2 is virtually impossible for a non-specialist to understand, at least as presented. It is not clear how many blood samples from the subject were analyzed, but it is possible that all the data described in this figure come from a single sample. My view is that a simple table showing enrichment or lack of enrichment of certain cell types would be easier to understand, along with a description of the number of samples analyzed for the subject with the mutation and some indication of the significance of the findings. Obviously, patient material may be limited, which is understandable, but it's important to convey how sure they are of these findings by detailing the numbers of replicates and, if possible, the significance of their differences with controls.
4. We appreciate the comparisons with the existing mouse models, but, based on the known domain structure, they are not perfect models of the human disease. I suspect the authors are trying to make the analogous mouse, but it might be helpful to have a short table or figure describing the differences in expected protein amount and sequence between the human point mutant and the available mouse mutants (and possibly the heterozygous Japanese subject), particularly in relation to the known domain structure of the protein.
5. Ideally, the data on cytokine secretion and mRNA turnover would be performed on cells from the analogous point mutant mice, but since those are not available, the authors had to rely on overexpression studies in HEK 293 cells, and a complex retroviral replacement strategy in cells from mutant mice. In my view, they should at least note the potential limitations of these approaches, in contrast to studies of cells in mice and/or humans with the exact mutations under

study in the endogenous protein.

6. I may have missed this, but it would be of interest to know if the authors had searched the available public human databases for this variant, presumably in heterozygous form, to know if this variant has been detected in the general population, at least as reflected in these databases.

Reviewer 1 major comments:

1. *Haploinsufficiency with milder phenotype?*

We agree with reviewer's 1 comments that the observed immunopathology in both heterozygous parents is suggestive of haploinsufficiency. Several lines of evidence support this observation:

- The available population data on Roquin-1 variants (<https://gnomad.broadinstitute.org>) reveals a very high intolerance for loss of function (pLI = 0,99) indicating that also heterozygous loss of function variants of RC3H1 are not tolerated in the human population.
- Our own experimental data further substantiate this statement; immunophenotyping analysis performed on PBMCs of the heterozygous R688* parents reveal an intermediate increase in ICOS expression. This was observed both on PBMCs and on immune cells derived from NOD-scid IL2 γ null (NSG) mice that were injected with PBMCs derived from the R688*/R688* patient or heterozygous parents (Fig.1A-B of this rebuttal). Furthermore, the sera of both parents display an intermediate increase in inflammatory cytokine levels (Fig.1C of this rebuttal). Similar observations can be made in murine models of Roquin-1 loss of function. Heterozygous carriers of Roquin-1 variants display intermediary levels of Roquin-1 targets such as Icos (Bertossi et al, JEM, 2011; Pratama et al, Immunity, 2013).
- To distinguish dominant negative behavior of the truncated R688* Roquin-1 fragment from haploinsufficiency, we supplemented murine WT T cells with the GFP tagged murine variant R687* Roquin-1 and compared it to Roquin-1 WT, Roquin-1 M199R and Roquin-1 AA1-509. The results of this experiment (Fig.1D of this rebuttal) suggest that the posttranscriptional activity of R687* Roquin-1 is target dependent. Whereas no effect on Icos expression could be observed by the addition of the R687* variant, Ox40 expression increased by the addition of both the Roquin-1 AA1-509 and the Roquin-1 R687* variant. This experiment indicates that the observed phenotype in the heterozygous carriers might be a consequence of deregulated posttranscriptional regulation in a target dependent manner. We envision future experiments to further investigate this in more detail (Fig.1D of this rebuttal).

Fig.1 Analysis of heterozygous R688* carriers.

Fig.1A

Fig.1B

Fig.1C

2. Figure 2: identity of cluster 12?

We have included a description of cluster 12 in this rebuttal (Fig.2). Given that the FlowSOM tree suggested that this cluster was closely resembling cluster 100 (Basophils), we used this cluster as a reference population (red histograms in Fig.2A). This analysis revealed that cluster 12 (black histograms in Fig.2A) is composed of poorly characterized events and that the used antibody panel was unable to clearly identify this population. Finally, examining the FSC/SSC gate of cluster 12 (black dot plot in Fig. 2A) revealed a pattern consistent with cellular debris rather than viable cells (cluster 100; basophils; red dot plot). This information was added in the results section of the manuscript and in Supp.Fig.2A of the revised manuscript. This is an apt illustration of the unbiased approach of FlowSOM compared to manual gating. In the latter approach these events would have been discarded early on in the analysis.

Furthermore, as requested by Reviewer1, the whole dataset is available on the public data base FlowRepository with repository ID FR-FCM-Z267. The R-code used for this analysis is accessible at <https://github.com/saeyslab/Roquin-1-hyperinflammation>.

Fig.2 Identity of cluster 12

Fig.2A

Fig.2B

3. *The reduced colocalization of Edc4 and RC3H1 R688* is at present not convincing. This requires additional experiments.*

We regret that the data supplied in the initial manuscript were not convincing to Reviewer 1. We want to point out that we performed a statistical analysis on >20 images confirming a reduced colocalization of the fluorescence specific to antibodies recognizing V5-R688* Roquin-1 and Edc4 (staining for P-bodies). We also stained the cells with an anti-eIF3 antibody to visualize stress granules and performed statistical analysis on >20 individual images. As the Reviewer 1 correctly notes, a speckled picture of V5-R688* is readily visible in Supp.Fig.5C, but we want to emphasize that this speckled nature of R688* Roquin-1 was only visualized in a setting of arsenite induced cellular stress and results in colocalization of RC3H1 with eIF3 (a marker of stress granules) rather than Edc4 (P-bodies). To further strengthen the dataset, we also visualized P-bodies with DCP-1 in HEK293T cells (Fig.3A of this rebuttal) and repeated the staining with eIF3 to visualize stress granules upon arsenite treatment (Fig.3B of this rebuttal). This experiment revealed that also colocalization of R688* Roquin-1 with this component of P-bodies but not stress granules is disrupted. These results are in line with our previous observations. Finally, upon request of Reviewer 3, we performed colocalization studies in primary T cells and found a similar reduced colocalization of R688* Roquin-1 and P-bodies (Fig. 20 in this rebuttal). We have added this information in the revised manuscript as Supp. Fig. 6A and B.

Fig.3. Colocalization of the R688* variant with Dcp1 but not eIF3 is impaired.

Fig.3A

Fig.3B

4. *Supplementary Figure1: Mepazine does not seem to inhibit degradation of R688* convincingly. Supplement with a densitometry.*

A very similar comment has been raised by Reviewer 3. To address this issue, we have further optimized the experimental conditions. Rather than using the more promiscuous compound phytohemagglutinin (PHA) to activate T cells, we used plate bound anti-CD3/anti-CD28 to expand T cells. Next, anti-CD3/anti-CD28 was removed to reduce Malt1 paracaspase activity and reduce Roquin-1 cleavage. Probing both WT and R688* Roquin-1 with anti-Roquin1 antibody (3F12) readily revealed both the full-length and truncated Roquin-1 (Fig.4 of this rebuttal). In these experiments, the pretreatment of the cells with the Malt1 inhibitor mepazine (20 μ M, 3h) effectively prevented the PMA/Ionomycin (1h) induced Roquin-1 cleavage. β -Tubulin is used as loading control (Fig. 4C and 4F). These new western blots have also been included in the revised manuscript as Supp. Fig. 1K. The experiment was replicated twice (Fig. 4A-F) with similar results. Due to the absence of a R688* Roquin-1 band in both replicate experiments upon PMA/Ionomycin treatment, a densitometry was not feasible.

Fig.4 Truncated R688* Roquin-1 is cleaved upon treatment with PMA/Ionomycin.

Fig.4A

5. In Figures 7G and 7I no statistics provided due to small sample size - it requires replication. Is this consistent with Supp. Figure 6D and 6E? please explain the discrepancy.

Additional replicates have been generated for Figures 7G and 7I. The data from three additional experiments have been added to the first 3 replicates already present in the original manuscript. To

reduce data spread due to day-to-day antibody staining variations, the expression of the Roquin-1 targets has been normalized to the expression of these targets in cells transduced with GFP fused wild type Roquin-1 (Fig.5 of this rebuttal). The additional replication allowed us to perform statistical analysis (one-way ANOVA with Tukey's test for multiple comparisons). This analysis confirmed that the R687* Roquin-1 variant was impaired to regulate the known Roquin-1 targets ICOS, Ox40, CTLA4 and resulted in increased TNF, IL-2 and IL-17A levels. This data resulted in new Figures 7B, D, E, G, I and J in the revised manuscript. In the original manuscript the secreted cytokine data (Supp. Figure 6D and 6E) were added to provide additional data about the role of the studied Roquin-1 variants in cytokine production. The transduction efficiency of the T cells with the various mutants ranged from 30% to 80% and was variable between constructs and experiments. As a consequence, the measured cytokine concentrations were the result of an inconsistent mixture of transduced cells containing the Roquin-1 variant and untransduced T cells devoid of both Roquin-1 and Roquin-2. This resulted in large inter and intra experimental variation. Given this uncontrollable variation, we decided to remove Supp. Figure 6D and 6E from the initial manuscript.

Fig.5 Dysregulation of Roquin-1 targets ICOS, Ox40 and CTLA4 and cytokines TNF, IL-2, IL-17A in presence of the R687* Roquin-1 variant.

Additional points

1. *The authors should carefully revisit the nomenclature where human and mouse genes/gene products are described.*

We have carefully checked gene/gene product nomenclature. Changes to the original nomenclature have been highlighted in red. We have addressed the gene/gene product names that were not in line with the HUGO nomenclature.

2. *18-year-old boy → inappropriate to be called boy, please revise.*

18-year-old boy has been altered into 18-year-old male adolescent. This text change is highlighted in red in the manuscript.

3. Please provide an illustration of all mutations in different domains of Roquin-1 described in mouse.

This comment is in line with reviewer 4 who also proposed to add the described pathogenic human Roquin-1 variants. The following illustration (Fig. 6) has been added to the manuscript as Supp. Fig. 8.

Fig.6 Graphical representation of transcripts and proteins encoded by the murine *Rc3h1* and human *RC3H1* variants.

4. When describing Figure 2C in the text, please specify NK cells as CD16+ or CD56+. This has been changed in the manuscript. Changes have been highlighted in red.

5. In FlowSOM analysis, it is suggested that manual gating did not identify reductions in naïve CD4 T cells as indicated by the clusters. If the number of clusters is reduced, does this issue persist? Given that reducing the number of clusters will result in the generation of fewer clusters with increased number of cells, this would result in a loss of resolution of the analysis. To avoid this loss of resolution, we opted to investigate the phenotype of cluster 154 in more detail and re-analyse this cell population by manual analysis. This revealed that the cell of cluster 154 (Red dot plot; Row1 of Fig. 7A) were primarily identified as CD3+ CD4+ CCR7+ CD45RA^{int} CD38^{lo} CD27+ PD1- cells, compared to naïve CD4 T cells present in clusters 153 and 154 (Red dot plots; CD45RA+ CD38^{hi}; Row2 and 3 of Fig. 7A). Applying a manual gating analysis on the expression of these specific markers, we could indeed find a reduced CD4 T cell population in the R688*/R688* patient compared to the healthy controls (Fig. 7B).

Fig.7 Manual re-analysis of cluster 154 (CD38^{lo} naïve T cells).

Fig.7A

Fig.7B

6. The aggregation of CD20/CD14 and CD3/CD14 cells is very interesting, please describe the markers expressed on each in these aggregates, do they differ between the patient and controls?

In Fig.8 of this rebuttal, a detailed phenotyping of the aggregates of CD20/CD14 (cluster 51) and CD3/CD14 (cluster 66) is provided. It can be observed that for the CD20/CD14 aggregates the phenotype is very similar between healthy controls and the R688*/R688* Roquin-1 patient (Fig.8A). In the setting of the CD3/14 aggregates the fluorescence of mainly the markers associated with T cells is lower in the case of the R688*/R688* proband (Fig.8B). The histograms of these markers (Fig.8C) reveal that the number of T cells involved in these aggregates is lower and account for the observed difference (Fig.8C). Finally, the additional table available in Fig.16 of this rebuttal reveals that the number, rather than the phenotype, of the aggregates is increased in the patient carrying the homozygous R688* variant. This is not surprising given that this was our primary research question when analyzing this dataset.

Fig.8 Phenotypical analysis of cluster 51 and cluster 66.

Fig.8A

Fig.8B

Fig.8C

7. In Figure 5K description, a “trend” is noted. Please remove this as it is misleading – the error bars are too large.

Upon request of reviewer 3, an additional experiment was performed to also provide cytokine levels upon ruxolitinib (RXL) treatment. This experiment was also used to obtain additional datapoints to reduce the spread in the data presented in Figure 5K of the original manuscript. A new Figure 5K is provided in the revised manuscript and is also presented in this rebuttal as Fig.9. As the reviewers can observe, RXL treatment significantly reduces the number of monocytes and eosinophils in the *sanroque* chimeras. The cytokine data presented as Fig.14A similarly demonstrates a reduction of TNF, CXCL9 concentration in the serum and CD64 expression of monocytes. In conclusion, these additional datasets confirm the inhibitory effect of RXL on the hypercytokinemia associated with the immune dysregulation phenotype in the setting of Roquin-1 dysfunction.

Fig. 9 Effects of Ruxolitinib on monocyte and granulocyte populations.

8. The Figure 6 description in the text is incorrect as the IP WB in 6F was clearly in 6A previously and it was not adjusted for in the text.

We apologize for this confusing reference in the text which was due to late adaptations to the submitted version. The wrong references in the text have been amended and have been highlighted in red.

9. In methods, mepazine was used in the range of 6.5uM-20uM. Please show that this concentration is sufficient - MALT1 WB or flow cytometry.

We would like to refer to figures 4A-F in this rebuttal for this specific comment. In these figures it can be observed that the pretreatment of T cells with mepazine (20µM, 3h) is sufficient to completely inhibit the formation of the Malt1 paracaspase activity dependent Roquin-1 cleavage product.

10. In Figure 1C, what is the band running at about 80kDa?

The presence of this additional band running at about 80kDa is indeed of interest as it is not present in the western blots displayed in the original report utilizing the anti-Roquin1-2 monoclonal antibody 3F12 that was also used in this study (Vogel et al; Immunity. 2013). Importantly, in this report only murine cells and tissues were probed with the 3F12 antibody. In the manuscript of Jeltsch et al (Nat. Immunol. 2014) an additional band running at approximately 75kDa could be observed in the lysates of human T cells. It was identified as an intermediate cleavage product as it completely disappeared upon MALT1 activation by PMA/Ionomycin stimulation.

First, we questioned whether an additional alternative transcript of Roquin-1 or Roquin-2 could be present in cells of human origin. The alternative splice products of Roquin-1 generated protein products containing 1125 or 1133 amino acids (Fig. 10A). Furthermore, the alternative splice products of Roquin-2 translate into proteins of 1191, 1064 or 506 amino acids (Fig. 10B). The alternative splice product of 506 amino acids would be recognized by the 3F12 clone but is similar in size as the MALT1 degradation product of Roquin-1 and Roquin-2 running at approximately 55kDa. Given these data, it is unlikely that the band running at 80kDa represents an alternative splice product.

A second well validated polyclonal antibody is available to probe Roquin-1 (Bethyl; A300-514A or Novus; NB100-655) which was raised against the immunogen that maps to the region 500-550AA of Roquin-1. Comparing both antibodies, we could observe that the band running at 80kDa detected by

the 3F12 clone (supplied by Millipore) was not detected by the polyclonal Bethyl antibody, which, however, cross-reacted with other bands below 80kDa. Furthermore, and in contrast to the study of Jeltsch et al, we could observe that the detected protein was not sensitive to MALT1 paracaspase activity as it did not disappear upon PMA/Ionomycin (in contrast to both Roquin-1 and Roquin-2; Fig. 10C). We similarly tested whether the secondary anti-rat antibody used in our studies could explain the presence of this band running at 80kDa. This experiment failed to visualize a band running at 80kDa.

Finally, we identified the peptide that is recognized by the monoclonal antibody by probing a synthesized peptide array, and we blasted the peptide sequence recognized by the 3F12 antibody in Roquin-1 (IELLPVNSALLQLV) and Roquin-2 (IDVLPVNFALLQLV) in the database (NCBI-blastp suite, homo sapiens). Filtering on proteins with a calculated molecular weight of 80kDa, we were able to identify TMEM63B as a possible candidate. Roquin-2 (VLPVNFALLQ) and TMEM63B (VLPVNFSGDLLE) share 8 out of 12 amino acids which might explain a low affinity binding to TMEM63B.

Fig. 10 Immunoblotting with clone 3F12 reveals a non-specific band running at 80kDa.

Fig. 10A

Name	Transcript ID	bp	Protein	Biotype	CCDS	UniProt	RefSeq Match	Flags
RC3H1-203	ENST00000367696.6	11261	1133aa	Protein coding	CCDS30940	Q5TC82	-	TSL:5 GENCODE basic APPRIS P3
RC3H1-201	ENST00000258349.8	10988	1133aa	Protein coding	CCDS30940	Q5TC82	-	TSL:1 GENCODE basic APPRIS P3
RC3H1-202	ENST00000367694.2	3775	1125aa	Protein coding	CCDS72987	Q5TC82	-	TSL:2 GENCODE basic APPRIS ALT2
RC3H1-204	ENST00000479099.2	481	No protein	Retained intron	-	-	-	TSL:3
RC3H1-205	ENST00000484867.1	368	No protein	Retained intron	-	-	-	TSL:3
RC3H1-206	ENST00000531594.1	303	No protein	Retained intron	-	-	-	TSL:3

Fig. 10B

Name	Transcript ID	bp	Protein	Biotype	CCDS	UniProt	RefSeq Match	Flags
RC3H2-203	ENST00000373670.5	9248	1191aa	Protein coding	CCDS43874	A0A024R899 Q9HBD1	-	TSL:5 GENCODE basic APPRIS P1
RC3H2-202	ENST00000357244.7	8964	1191aa	Protein coding	CCDS43874	A0A024R899 Q9HBD1	NM_001100588.3	TSL:5 GENCODE basic APPRIS P1 MANE Select v0.5
RC3H2-205	ENST00000423239.6	3623	1064aa	Protein coding	CCDS48014	Q9HBD1	-	TSL:5 GENCODE basic
RC3H2-201	ENST00000335387.9	3081	506aa	Protein coding	CCDS87685	A6NHN2	-	TSL:1 GENCODE basic
RC3H2-206	ENST00000454740.5	923	188aa	Protein coding	-	H0Y5D9	-	CDS 5' incomplete TSL:5
RC3H2-207	ENST00000471874.2	749	218aa	Protein coding	-	Q9HBD1	-	TSL:1 GENCODE basic
RC3H2-204	ENST00000398671.2	666	111aa	Protein coding	-	Q4VXB2	-	CDS 5' incomplete TSL:5
RC3H2-210	ENST00000498479.5	5746	478aa	Nonsense mediated decay	-	Q9HBD1	-	TSL:2
RC3H2-208	ENST00000478216.1	547	No protein	Processed transcript	-	-	-	TSL:3
RC3H2-209	ENST00000495727.2	420	No protein	Retained intron	-	-	-	TSL:2

Fig. 10C

T cell lymphoblasts

11. In Figure 1, please include Sanger sequencing result of the patient and parents to show segregation of the signal.

Sanger sequencing results of a healthy control, the R688*/R688* patient and the heterozygote father have been included in the revised manuscript as a new Fig.1B. The results are also shown in this rebuttal as Fig. 11.

Fig. 11 Sanger sequencing of the R688*/R688* proband, first degree relative and healthy control.
c.2026C>T

12. In Figure 3, please include other Roquin-1 targets apart from TNF (ex IL6, IRF4). Also, why does TNF expression at the transcript level appear the same between control and patient?

Using RT-qPCR, we have assessed the expression levels of additional Roquin-1 targets in primary T cells expanded with anti-CD3/CD28 and IL-2. The results of this experiment have been included in the revised manuscript as Supp.Fig.3C and Fig.12 in this rebuttal. These data revealed the upregulation of a number of validated Roquin-1 targets. We also assessed the gene expression of IL-6 (a known Regnase-1 target) and found it similarly upregulated. A possible explanation for this observation could be the cooperative activity of both Roquin-1 and Regnase (Jeltsch et al, Nat Immunol, 2014). This would require additional experiments and remains beyond the scope of this manuscript.

As Roquin-1 acts as a posttranscriptional regulator, loss of function variants of Roquin-1 will not alter the induction of TNF transcripts upon PMA/Ionomycin. Furthermore, as Roquin-1 is degraded upon TCR stimulation (mimicked by PMA/Ionomycin) (Jeltsch et al, Nat Immunol, 2014), TNF transcripts will not be degraded by Roquin-1 shortly after stimulation (60' PMA/Ionomycin). Hence, similar TNF levels in both the patient and healthy controls are to be expected. In contrast, mepazine pretreatment prevents the MALT1 dependent cleavage of Roquin-1 and as a consequence promotes TNF transcript degradation. This reduction of transcript degradation in the R688* cells indicates the inability of mepazine stabilized R688* Roquin-1 variants to induce TNF transcript degradation.

Fig.12. Relative quantification of posttranscriptionally regulated mRNA transcripts in the R688*/R688* proband.

13. In Figure 4J, the values for cytokines are expressed in OD. Were appropriate standard curves used? Please express this as pg/mL.

Due to technical issues, one of the standard curves (IL-10) was of insufficient quality to be used to calculate cytokine concentrations. To amend this issue, the experiment has been repeated and the new data can be found as Fig. 4J in the adjusted manuscript.

14. In Figure 4K, significance bars are duplicated, please amend.

We thank the reviewer for the detailed examination of this figure. We have amended the figure and can be found as a new Fig. 4K in the revised manuscript.

15. In Figure 6F, the *edc4* band of IgG control seems to have a smear with a band coming up right after the cut. Please do not cut that out/please show the full WB.

As requested, an unedited western blot (Fig.13) is supplied to allow reviewer 1 analyze the immunoprecipitation experiment in more detail. These images have also been uploaded in the source data file as required by Nature Communication policy.

Fig.13 Impaired co-immunoprecipitation of the R688* Roquin-1 variant with CNOT1.

16. In Supp. Figure 6D-F, please provide pg/mL concentration (not normalized).

As detailed in major comment 5 of Reviewer 1, Supp. Figure 6D-F have been removed from the adjusted manuscript.

Reviewer 2 comments

1. Some HLH-associated data are missing for the index patient. If there is stored plasma, are the authors able to report IL18, IL18-bp, CXCL9 levels?

We thank Reviewer 2 for providing his valuable insights how to extend the characterization of the hyperinflammatory disease associated with Roquin-1 loss-of-function. These biomarkers are indeed helpful to further elucidate the pathophysiology of the observed hyperinflammatory syndrome. Whereas high total IL-18 (+24.000pg/ml) performs well as a diagnostic biomarker to distinguish Macrophage Activating Syndrome (MAS) from Hemophagocytic Lymphohistiocytosis (HLH), generating the ratio of total IL-18 and CXCL9 further promotes the distinction between both hyperferritinemic disorders (Weiss et al, Blood, 2018). Although IL-18BP similarly was increased in MAS compared to HLH and mainly free IL-18 determines susceptibility to more severe TLR9-induced MAS (Weiss et al, Blood, 2018), IL-18BP quantification in our opinion does not really add additional information.

In the new Fig.3A in the revised manuscript and Fig.14A in this rebuttal, we provide the details of increased IL-18 and CXCL9 in our patient. As can be observed, IL-18 is consistently upregulated in our patient (+/- 200pg/ml) but this is significantly below the lower threshold that is suggested by Weiss et al. to be suspicious of MAS. Measuring CXCL9 by ELISA revealed an increase of this cytokine in the serum of the R688*/R688* proband (revised Fig.3A and Fig.14A in this rebuttal) and the calculated ratio of IL-18 over CXCL9 (revised Supp. Fig3A and Fig.14B in this rebuttal) is similarly supporting the hypothesis that the clinical phenotype of the proband is more reminiscent of HLH. Indeed, when comparing the published ratio for MAS and HLH (Fig.14C in this rebuttal – adapted from Weiss et al,

Blood, 2018), the IL-18 over CXCL9 ratio in our proband is compatible with the calculated ratios in the setting of HLH.

Fig.14 Analysis of IL-18 and CXCL9 levels in the serum of the R688*/R688* proband.

Fig.14A

Fig.14B

Fig.14C

2. Outcome of the patient is not described. Given the mouse response, a trial of ruxolitinib or similar agent seems reasonable. Patient response to such an intervention would greatly increase the scientific and clinical impact of this study. Did this patient undergo stem cell transplant?

Reviewer 2 indeed highlights an important issue. Currently, the patient is clinically stable under chronic treatment with cyclosporin A (CSA) and therefore, a bone marrow transplant was not yet suggested. But considering the considerable toxicity of long-term treatment with CSA and our evidence suggesting ongoing subclinical inflammation with persistent increased cytokines under

therapeutic doses of CSA, addition of/switching to alternative immunosuppressive medication seems warranted. At this stage, we do not have yet obtained the required approvals for the off-label use of ruxolitinib in this particular setting. Nonetheless, Novartis has been contacted to discuss the possibility of compassionate use of JAKAVI (ruxolitinib). Also, in the advent of a poor control during a next relapse of a hyperinflammatory episode, the use of JAKAVI as a rescue treatment could be envisioned. In this regard, it is of interest to note that a clinical HLH trial testing the value of ruxolitinib (NCT02400463) is ongoing. The outcomes of this trial will be of interest to guide the therapeutical approach in our patient.

3. In Figure 5, response of immune cell populations to JAK1/2 inhibition is described. It would be informative to also include cytokine responses.

On request of Reviewer 2, additional experiments with ruxolitinib have been performed to measure cytokine production. We opted to measure IFN γ , TNF and CXCL9 (IFN γ responsive gene) and quantify CD64 (IFN γ response gene) median fluorescence on the surface of monocytes. This experiment revealed the significant reduction in TNF, CXCL9 and CD64 whereas IFN γ remained unchanged (Fig.15A of this rebuttal). These results fit with the concept that IFN γ is released by activated T cells due to Roquin-1 dysfunction and ruxolitinib acts downstream of the IFN γ receptor (JAK1/JAK2 dimers) to inhibit subsequent cytokine release. We also interrogated whether ruxolitinib treatment would reduce effector CD4 and CD8 T cells. In line with the hypothesis that T cell activation is due to cell intrinsic effects of Roquin-1 dysfunction, the percentage of activated T cells was not reduced (Fig.15B of this rebuttal). The results of this experiment have been added to the revised manuscript as Fig. 5L and Supp. Fig.5B.

Fig.15 Effects of Ruxolitinib in *sanroque* mice on the cytokine levels IFN γ , TNF, CXCL9, the cell surface marker CD64 and effector T cells.

Reviewer 3 comments

Tavernier et al describe an interesting observation based on a single patient carrying a homozygous point mutation (R688*) of human Roquin that results in the expression of a C-terminally shortened form of Roquin. This patient suffers from periodic hyperinflammation and the authors have performed multiple experiments to link this clinical phenotype to the mutant Roquin protein. There is extensive knowledge defining Roquin as a (negative) posttranscriptional regulator of inflammatory mRNAs. Hence, the study was designed to (i) phenotype the immunopathological syndrome of the patient at high resolution by assessing immune cell subpopulations and by determining changes within the cytokine milieu ex vivo and in vitro, (ii) relate this to the phenotype of an already described mutant Roquin mouse strain carrying another point mutation (M199R), and (iii) establish a mechanism for the

R688* Roquin mutant by assessing its subcellular localization and interaction with EDC4 and CCR4. Despite these efforts, the data sets remain descriptive and mostly do not reveal direct effects or a mechanism. Key experiments such as reverting the mutation in patient-derived cells by genomic engineering to proof that increased cytokine levels become normalized are missing. Reconstitution of murine Roquin-deficient T-cells with R688* variants fails to show a function of this mutant for TNF α , IL-17a and IL-2 expression. No experiments were performed to proof that R688* affects mRNA deadenylation or mRNA decay in model systems (such as HEK293 cells used in Fig. 6) or even in immune cells. A confounding problem is that in many experiments multiple healthy controls were evaluated against a single sample only which was available from the patient. Furthermore, comparing the data presented in Fig.2 versus Fig. 3A, an alternative explanation for the patient's clinical symptoms would be that there are defects in the myeloid system that result in periodic release of all major proinflammatory cytokines to the systemic circulation and this then causes long-term secondary differentiation defects in immune cell subpopulations.

Specific comments:

Fig.S1J: the protein band intensities of full length Roquin-1, R688 Roquin-1 and the cleaved Roquin fragment need to be quantified from replicate immunoblot experiments. As shown, the proposed increase in cleaved Roquin forms (including R688*) upon PMA/Ionomycin treatment and the inhibitory effects of Mepazine are not particularly obvious and seem to be inconsistent between conditions.*

As requested by both reviewer 1 and 3, replicate western blots using an optimized protocol have been generated. We kindly refer reviewer 3 to the rebuttal to the fourth comment of reviewer 1 and to Fig. 4A-F of this rebuttal. The new data have also been included in the revised manuscript as Supp. Fig. 1K.

Fig.2, Fig. S2. These two figures are hard to follow for non-immunologists not familiar with all the different immune cell populations that can be defined by sets of surface markers. Multiple subsets of immune cells identified by FACS analysis are discussed in the text but are not shown in the figures (e.g. clusters 194, 145, 91). The data reveal very complex shifts in the T-, B-, monocyte and NK cell repertoire. However, no functional analysis is provided to demonstrate that these immune cells have indeed altered cytokine profiles or cytotoxic activities or other loss- or gain-of-function phenotypes. These data need to be restructured and focused on the strongest differences between healthy controls and the patient.

We regret the confusion generated by Fig.2 and Supp. Fig.2. We have amended these figures with respect to both the comments of reviewer 3 and 4. To address the raised issues, we have modified the figures in several aspects:

- We have concatenated the clusters that have been mentioned in the text but not shown in the figures (e.g. clusters 194, 145, 91) with the similar clusters from which the phenotype is given in the figures. The phenotypic analysis derived from these concatenated files does not differ from the original analysis and resulted in the adapted Fig.2D and 2E. The adjusted panels can be viewed in Fig.16 of this rebuttal.
- Functional analysis (cytokine production and/or cytotoxic activity) have been performed but was not explicitly mentioned in the results section of the original manuscript. For instance, cytokine production and lytic capacity of NK cells have been tested in the initial clinical work-up of the patient and was found normal. This data is mentioned in Table1 of the revised manuscript. Furthermore, cytokine production by both CD4 and CD8 T cells has been assessed by ex-vivo PMA/Ionomycin stimulation in presence of BrefeldinA. This data is now included in a new Supp.Fig.2B (Fig. 17 of this rebuttal). Furthermore, the result section has been restructured to the comments of reviewer 3 and focuses on the strongest differences.
- On behalf of reviewer 3 and 4, we have added Supp.Fig.2A and Supp.Fig.2B in the revised manuscript (Fig. 17 of this rebuttal). These figures/tables give more insight in the changes of the immune cell clusters on the one hand (Supp.Fig.2A) and promote the comparison with the traditional manual analysis on the other hand (Supp.Fig.2B).

- Along similar lines, Supp. Fig.2B will allow the readers to observe that the observed phenotype of the patient is relatively stable given that replicating the data at different ages reveals a similar phenotype. This strengthens the data and we thank reviewer 4 for pointing this out.

Fig.16 Unbiased clustering of immune cell populations in the R688*/R688* proband using FlowSOM

Fig.17 Boxplot analysis of differentially expressed clusters and manual analysis of immune cell populations in the R688*/R688* PBMCs.

Supplementary Figure 2

The basis of the in-silico approach to cluster flow cytometry data needs to be explained for non-specialists. By which criteria were some sorted populations manually annotated?

The specific details on the basis of the unbiased clustering algorithm FlowSOM have been described in detail in the articles Van Gassen et al. Cyto A. 2015 and Saeys et al. Nat Rev Immunol. 2016. In short, the FlowSOM algorithm is both a clustering and visualization tool for flow cytometry data, using self organizing maps (SOM) to reduce dimensionality and clustering of the data. Upon generation of the SOM, the nodes are organized in a minimal spanning tree (MST). Nodes containing cell populations that are the most similar will be connected, generating a branched, acyclic graph. We made use of this clustering tool to ensure we did not overlook any populations of interest in the manual gating. In contrast to other clustering tools, FlowSOM typically uses a larger number of nodes than the expected cell types, providing additional information concerning cell populations (activation/differentiation state, immune cell subsets, etc.) that otherwise would have been missed. To help the scientist in

interpreting these generated nodes an additional clustering of the node centers is performed ('metaclustering'). Finally, the identity of well-known populations was determined based on the manual gating of key surface molecules expressed by these immune cell populations (e.g. CD14 for monocytes). A concise version of this explanation has been introduced in the results section of the revised manuscript and is marked in red.

How can be ruled out that the changes have developed slowly over time (or due to immunosuppressive treatment by CSA) and are thus not directly related to the R688 mutation?*

Reviewer 3 raises here an important issue that is not only of interest for this particular case. The phenotypic alterations of the immune system of many if not all patients suffering from primary immune deficiencies are the consequence of a complex interaction of genetic defects, chronic (immunomodulatory) medication and ongoing infections. The reviewer is right to point out that chronic CSA treatment has profound effects on the lymphoid immune system. These effects have been described extensively in literature and we provide a comprehensive summary in table 1 of this rebuttal. From this analysis it is clear that CSA affects particular aspects of both T and B cell biology, including B and T cell proliferation and differentiation, surface molecule expression and cytokine production. Given that we observe an increased expression of ICOS and OX40 on T cells, an increased number of B cells, an accumulation of (exhausted) effector T cells and enhanced cytokine production by both CD4 and CD8 T cells, we consider that it is unlikely that these alterations are the consequence of long term CSA treatment. Nonetheless, we are unable to completely rule out that the observed defects in B cell maturation are a consequence of chronic CSA treatment. This comment has also been added in the revised manuscript and is highlighted in red. Finally, a strong analogy can be drawn between the immunophenotype of the R688*/R688* Roquin-1 proband and the published roquin-1 transgenic mice. Indeed, both display increased numbers of regulatory T cells, effector CD4+ and CD8+ T cells, myeloid cells and cytokine levels. Altogether, these data make a strong case that the observed immunophenotype is mainly a consequence of the loss of function variant of Roquin-1 rather than the chronic CSA treatment.

Table 1. Effects of calcineurin inhibition on the human immune system.

T cells		
CD4 T cells	Reduced	Bueno O. et al. PNAS. 2002
Tregs	Reduced	Kim H. et al. Immunology. 2013; Abadja F. et al. Transplantation. 2011
Effector cells	Unaffected	Kim H. et al. Immunology. 2013; Brooks EG. et al. Clin Immunol Immunopathol. 1993
CD8 T cells	Reduced	Bueno O. et al. PNAS. 2002
Effector cells	Unaffected	Kim H. et al. Immunology. 2013
Proliferation	Reduced	Kasaian et al M. J.Immunol. 1990
Cytokine production		Brooks EG. et al. Clin Immunol Immunopathol. 1993
IL-2	Reduced	Kasaian M. et al. J.Immunol. 1990; Brooks EG. et al. Clin Immunol Immunopathol. 1993; Bueno O. et al. PNAS. 2002
IFNg	Reduced	Heidt S. et al. Clin Exp Immunol. 2010
IL-21	Reduced	Abadja F. et al. Transplantation. 2011; De Bruyne R. et al. Clin Exp Immunol. 2015
Surface molecules		
HLA-DR	Reduced	Brooks EG. et al. Clin Immunol Immunopathol. 1993
CD25	Reduced	Brooks EG. et al. Clin Immunol Immunopathol. 1993
CD69	Reduced	Heidt S. et al. Clin Exp Immunol. 2010
ICOS	Reduced	Heidt S. et al. Clin Exp Immunol. 2010
CTLA-4	Reduced	Heidt S. et al. Clin Exp Immunol. 2010
Cytolytic activity	Reduced	Brooks EG. et al. Clin Immunol Immunopathol. 1993
NK cells		
Activation	Unaffected	Kasaian et al. J.Immunol. 1990
B cells		
Plasma blast differentiation	Unaffected	De Bruyne R. et al. Clin Exp Immunol. 2015
Naive B cell proliferation	Reduced	De Bruyne R. et al. Clin Exp Immunol. 2015
Class switching	Reduced	De Bruyne R. et al. Clin Exp Immunol. 2015
Immunoglobulin production	Unaffected	Heidt S. et al. Transplantation. 2008

Fig. 3A: There are profound increases in serum levels of IL-1b but also the IL-1R antagonist (IL-1RA). Production of IL-1b alone would be sufficient to cause a hyperinflammatory syndrome, but this possibility is not discussed at all. Also how an altered balance of IL-1beta and IL-1RA could contribute to the disease phenotype is not discussed. Similarly, arguments apply for elevated serum levels of IL-18. In

the light of these observations, the reason why the authors focus largely on adaptive immune cells remains unclear.

We thank Reviewer 3 for highlighting the potential role of the myeloid compartment in the immune dysregulation syndrome of the R688*/R688* patient. In the revised manuscript we discuss the role of the myeloid compartment in more detail. These adjustments have been highlighted in red. In this rebuttal, we summarize the main reasons why we have focused on the role of the adaptive immune cells.

Careful interpretation of both our and published data might suggest a role for Roquin-1 in the myeloid cell compartment in the pathogenesis of the immune dysregulation of the R688*/R688* patient:

- Ongoing presence of elevated cytokines in the serum despite therapeutic CSA concentrations suggest an alternative cellular source of cytokine production. The release of IL-1 and IL-18 could be myeloid cell derived although at present it is not clear whether also for instance non immune cells such as epithelial cells contribute to disease. It is of importance to note that although we are able to measure elevated levels of cytokines in the serum of the R688*/R688* patient, the clinical symptoms of hyperinflammation are controlled by cyclosporinA treatment. This suggests that inhibition of NFAT, an important regulator of T cell activation is sufficient to suppress the hyperactivation of the (adaptive) immune system.
- Rag1^{-/-} Rc3h1^{san/san} mice are susceptible to both LPS induced lethality and arthritis in an adoptive model of rheumatoid arthritis (Pratama et al, Immunity, 2013), suggesting a role for Roquin-1 in other cell types beyond B and T cells. These pathologies are mainly driven by TNF which we report to be similarly increased in the supernatant of both T cells and monocytes derived of the R688* patient.

To address whether also IL-1 β and IL-18 secretion is increased in R688*/R688* monocytes, we stimulated monocytes from HCs and the R688* patient with LPS and/or ATP. The results of these experiments can be observed in Fig.18 of this rebuttal and also in Supp.Fig.3B of the revised manuscript. These experiments were performed twice and gave consistent results. In contrast to the increased cytokine release of both TNF and IL-6 upon LPS treatment of R688*/R688* monocytes, IL-1 β and IL-18 were not significantly increased upon ATP or LPS+ATP treatment. We observe a trend towards higher concentration of IL-18 upon stimulation of R688*/R688* monocytes but this is minor compared to TNF and is not suggestive for a selective activation of the inflammasome in the presence of the R688* Roquin-1 variant. * = p<0,05; ***= 0,001< p <0,0001 (unpaired T-test).

Fig.18 Cytokine levels of TNF, IL-6, IL-1 β and IL-18 in the supernatant of stimulated R688*/R688* monocytes.

A central paradigm in the pathogenesis of hyperinflammatory syndromes is the hyperactivation of T cells and macrophage stimulation causing a cytokine storm resulting in severe, sometimes deadly immunopathology. Hemophagocytic LymphoHistiocytosis (HLH) and Macrophage Activation Disorder (MAS) represent two distinct clinical entities among the spectrum of hyperinflammatory disorders. Both syndromes culminate in a convergent pathway in which IFN γ release by overactivated T cells drives macrophage activation. Nonetheless, the initial immunological and/or genetic disorder

underlying the hyperinflammation are very different (Schulert et al, 2018). Familial HLH is driven by genetic defects that impair the perforin-dependent lymphocyte cytotoxic function, resulting in the accumulation of activated T cells. In MAS, the genetic causes underlying disease are more diffuse, aside of hypomorphic variants in the perforin pathway also genetic defects resulting in autoinflammation have been described (NLRC4, XIAP). Whereas in HLH the myeloid cell activation is most likely secondary to hyperactivation of T cells and therapy has focused on the quiescing of the adaptive immune response (e.g. calcineurin inhibitors such as cyclosporin A; alemtuzumab); targeting the myeloid compartment can be successful in selective cases of MAS (Schulert et al, 2018). Indeed, treatment with IL-1RA (anakinra) has been successful in MAS secondary to systemic onset juvenile idiopathic arthritis (Sönmez HE. Et al. Clin Rheumatol. 2018). Along similar lines, a clinical trial using recombinant IL-18BP (Tadekinig α) in the setting of MAS secondary to autoinflammatory disease caused by underlying genetic defects in NLRC4 and XIAP is currently ongoing (Jordan M. Blood. 2018).

In this particular case, the initial clinical, functional and genetic diagnostic work-up was not suspect for an underlying autoinflammatory disorder or rheumatological immune disease. Indeed, the patient did not present with rheumatic disease or recurrent fevers; we were unable to identify increased levels of autoantibodies; and whole exome sequencing did not identify pathogenic variants associated with autoinflammatory disorders.

Furthermore, a recent publication of Weiss et al. identified IL-18 and CXCL9 as useful biomarkers to distinguish between HLH and MAS (Weiss et al. Blood. 2018). As can be observed in Fig. 14A-C of this rebuttal, the levels of IL-18 (200pg/ml vs 24.000pg/ml) and the ratio of IL-18 over CXCL9 are indicative of an HLH-like syndrome rather than MAS (adapted in revised Fig. 3A and revised Supp. Fig. 3A)

Our decision was also based on the large body of evidence highlighting the important immunoregulatory role of Roquin-1 in T cells. Loss of Roquin-1 and Roquin-2 in T cells results in severe immunopathology characterized by expansion of effector T cells, lymphocytic infiltration of organs and the increased secretion of a number of cytokines (IL-6, TNF, IL-17A and IL-10) (Vogel et al, Immunity, 2013; Jeltsch et al, Nat Immunol, 2015). Furthermore, comparing the phenotype of the Roquin-1^{fl/fl} crossed to the Vav-CRE with the Roquin-1^{fl/fl} crossed to the CD4-CRE revealed only minor differences suggesting that the majority of phenotypic alterations are driven by the adaptive immune system (Bertossi et al, JEM, 2011). Analysis of the *sanroque* mice also revealed that IFN γ is one of the main drivers of the disease phenotype. Indeed, the most salient features of immunopathology are absent in *sanroque* mice lacking the receptor for IFN γ (Lee et al, Immunity, 2012).

In conclusion, these findings and additional above-mentioned experiments provide further arguments that the adaptive immune system rather than myeloid cells is involved in the pathogenesis of immune dysregulation in presence of the hypomorphic R688* Roquin-1 variant.

Fig. 3B: related to this point: activation of T-cells in vitro by artificial addition of PHA/IL-1 only doubles expression of TNFa/IFNg, whereas in the blood samples changes of cytokines mainly released from the myeloid compartment increase at the log scale!

In our opinion, the comparison of the datasets presented in Fig.3A and Fig.3B is not appropriate and does not provide evidence for a primary role of the myeloid compartment in the development of disease.

- Whereas Fig.3A depicts in-vivo serum cytokine levels of 'non-inflamed' healthy controls compared to the 'hyper inflamed' R688*/R688* proband, the dataset in Fig.3B directly compares cytokine levels in T cells and monocytes of both healthy controls and R688*/R688* proband upon in-vitro stimulation with PMA/Ionomycin or LPS. As such, the data in Fig.3A is derived from cells in very different states of activation and Fig.3B is examining cytokine release by cells that have been exposed to similar amounts of stimulatory agents. Hence, the direct comparison of differences in cytokines release by stimulated cells derived from healthy controls and the R688*/R688* patients with the serum cytokines of non-inflamed healthy controls and hyper-inflamed R688*/R688* is not advisable.

- The in-vivo data presented in Fig.3A reflects the complex interactions of multiple immune and non-immune cell types releasing a plethora of pro-inflammatory and regulatory effector molecules ultimately resulting in numerous positive and negative feedback loops. As such, the observed increases of myeloid derived cytokines can be indirect and are not sufficient as an argument for a disease driven by the myeloid compartment. Indeed, strong increases of myeloid derived cytokines are also evident in the serum of patients suffering from diseases that are primarily driven by adaptive immune cells (e.g. graft versus host disease, autoimmune disease, cytokine release syndrome upon CAR-T cell therapy).

Fig. 3E: this figure shows inducible mRNA steady state levels of TNF α in both, the healthy controls and the patient samples. The suppressive effect of Mepazine on controls cannot be interpreted as altered mRNA stability as discussed in the text unless results from mRNA decay assays are provided. The mRNA steady state levels for TNF and IFN γ mainly show comparable activation in the patient samples and weak (if at all) or variable effects after Mepazine treatment.

In accordance with reviewer 1 and 3 comments, we have tried to strengthen the claim that the hypomorphic R688*/R688* Roquin-1 variant encodes a protein with reduced posttranscriptional activity. To address this question, following experiments have been performed:

- Quantification of mRNA transcripts of known Roquin-1 targets in patient derived T cells (Supp.Fig.3C in the revised manuscript and Fig.12 in this rebuttal).
- Kinetics of ICOS mRNA decay upon actinomycin D treatment in patient derived T cells (Fig.6G in the revised manuscript and Fig.19 in this rebuttal).
- Assessment of PolyA tail of *Icos* mRNA in murine T cells transduced with WT or R687* Roquin-1 (Fig. 6H in the revised manuscript and Fig. 20 in this rebuttal).

As can be seen from the chase experiments performed with Actinomycin D (Fig.19A), the decay rates of ICOS mRNA are significantly delayed in the R688*/R688* T cells compared to healthy controls. The figure represents pooled data from 3 independent experiments. We have also performed the kinetics for other Roquin-1 target genes such as *TNF*. The decay rates for this gene showed similar trends in each separate experiment, but the variation between experiments was too big to pool the data and perform statistical analysis. We speculate that this is due to technical issues. Indeed, to avoid Malt1 dependent cleavage of Roquin-1, these experiments were performed with unstimulated T cells and the *TNF* transcripts are hence expressed at low levels, probably beclouding proper analysis. For this reason, we opted to show only the mRNA decay analysis of *ICOS* in the revised manuscript.

Fig.19. ICOS mRNA is stabilized in presence of the R688* Roquin-1 variant.

Fig. 4/5: The necessity to include the extensive data set from the sanroque (M199R) mice can be questioned. Multiple parameters do not really match the patient's characteristics and the authors eventually conclude that the reduced Roquin function in these mice result in a more pronounced MAS-like phenotype; hence altogether these mice simply have a different immune pathology. Along the same lines: the bone-marrow transplantation and JAK1/2 inhibitor experiments are interesting but do not help to unravel the R688 phenotype.*

We agree with the reviewer 3 that the *sanroque* mice have their limitations as a model for the R688*/R688* patient. Nonetheless, the comparison of the phenotype of the mouse and human

immune system in presence of hypomorphic Roquin-1 variants (M199R vs R688*) reveals a number of important parallels.

- Immunophenotyping reveals a striking resemblance across species.
- Compared to wild types, the mice develop a more severe hyperinflammation upon TLR challenge. Although one can dispute the choice for this disease model, the alternative models for hyperinflammation were not representative (perforin KO mice) for the observed disease or would require extensive breedings (Mouse Cytomegaly virus infections in mice on a Balb/C background).

Based on the data generated by transducing murine T cells with the different Roquin-1 variants (Fig.5 of this rebuttal), we speculate that the limitations to use the *sanroque* mice as a model for the disease observed in the R688*/R688* proband might be due to the remaining posttranscriptional activity in the M199R (*sanroque*) variant. Indeed, comparing both variants reveals that the R688* mutation results in more severe dysregulation in murine T cells compared to the M199R mutation.

Fig. 6: At this point of the story, a turn is made to reveal a mechanism for R688 Roquin. Fig. 6A is not mentioned in the text. Fig. 6A-E show altered distribution of Roquin to EDC4-positive granules. No other markers were included to prove that these are indeed P-bodies. P-bodies are now largely viewed as storage sites for mRNAs stalled in translation rather than as sites of specific mRNA decay (see Hubstenberger, A., et al., P-Body Purification Reveals the Condensation of Repressed mRNA Regulons. Mol Cell, 2017. 68(1): p. 144-157). Therefore, the altered P-body localization of R688* Roquin is not indicative of altered mRNA decay functions. Why were these experiments only performed in HEK293 cells and with overexpressed proteins?*

Reviewer 3 raises similar concerns as reviewer 1 concerning P-body localization of Roquin-1. To address these issues, we have performed following experiments:

- Colocalization studies in HEK293T cells of transfected WT or R688* Roquin-1 and a second P-body marker DCP1 (Sup. Fig. 6B in the revised manuscript and Fig. 3 in this rebuttal).
- Colocalization studies in murine T cells transduced with WT or R687* Roquin-1 and a third P-body marker RCK (Sup. Fig. 6C in the revised manuscript and Fig. 20 in this rebuttal).

The results of these experiments are in line with the colocalization studies with Edc4 and reveal a reduced presence of mutant Roquin-1 in P-bodies. We thank reviewer 3 to clarify the current views on P-body function and the fate of P-body localized mRNAs. We have changed the manuscripts accordingly. These changes have been highlighted in red. Finally, we also performed the experiments in murine T cells, revealing a similar defect for the mouse R687* Roquin-1 variant.

Fig.20. Reduced colocalization of GFP- R687* Roquin-1 with Rck in murine T cells.

Fig. 6F: The observation of loss of the CNOT1 subunit from R688 Roquin immune complexes is important. Does this R688* Roquin complex has less deadenylase or decapping activity? Clarifying this point is crucial and several assays are available to determine polyA tail length of model transcripts or endogenous mRNAs.*

To assess the impact of the R688* mutation on the posttranscriptional function of Roquin-1, we studied the poly(A) tail of *Icos* mRNA in murine T cells transduced with WT or R687* Roquin-1. In short, naïve CD4+ T cells derived of Rc3h1-2^{fl/fl}; CD4-ERT2; rtTA CRE mice were purified and treated with tamoxifen for 24 hours. After 40 hours of anti-CD3/CD28 activation, T cells were retrovirally transduced with a vector encoding doxycycline inducible GFP-fused WT or R687* Roquin-1. T cells were expanded and after 6 hours stimulation with doxycycline, GFP positive cells were sorted using FACS. Subsequently, T cells were lysed and RNA extracted. The poly(A) was assessed using the Poly(A) Tail-Length Assay Kit (Thermofisher, 764551KT) following manufacturer's instructions. This experiment was repeated twice and representative data is shown in Fig.6H in the revised manuscript and Fig.21A of this rebuttal. From this figure it can be observed that in the absence of Roquin-1, an accumulation of *Icos* mRNA with variable poly(A)tail length occurs (visible as a clear smear around approximately 300bp). Upon transduction with a vector encoding WT Roquin-1, this accumulation is strongly reduced indicating that WT Roquin-1 reduces the pool of *Icos* mRNA with a poly(A) tail. In contrast, the introduction of the R687* Roquin-1 only slightly reduces this smear of poly(A) tailed *Icos*. This was quantified by generating a ratio of poly(A) *Icos* over *Icos* mRNA running at approximately 200bp. As a control, *Icos* specific primers were designed consisting of a forward primer that was also used to assess poly(A) tail length and a reverse primer that binds in the 3' end of the coding sequence. In contrast to *Icos*, the results for *Tnf* and *Nfkbid* were more difficult to interpret (Fig.21B,C). Although a poly(A) tailed fraction of *Tnf* and *Nfkbid* was clearly visible in absence of Roquin-1, no or only few poly(A) tailed *Tnf* could be observed in presence of both the WT and the R687* Roquin-1 variant. Moreover, no difference could be seen comparing WT and R687* Roquin-1. Several explanations could be envisioned. First of all, this could be due to technical limitations. *Tnf* was assessed in absence of T cell stimulation to prevent Roquin-1 cleavage by Malt1. The absence of T cell stimulation results in reduced amounts of *Tnf* transcripts and a failure of this technique to reveal a difference between WT and R687* Roquin-1 in poly(A) tail length in the case of *Tnf*. Furthermore, although we collected cells after 6 hours of doxycycline administration, the absence of difference between both variants could be due to supraphysiological expression of both variants. Another possible explanation can be found in the differential regulation of *Icos* and *Tnf*. Whereas *Icos* is regulated by Roquin-1 and Regnase-1 and miR-146a the mRNA of *Tnf* may rather be a target of Roquin-1 and Tristetraprolin. The presence and redundant or cooperative functions of these additional posttranscriptional regulators is a possible explanation for the lack of difference between both variants. Finally, it is clear from literature that Roquin-1 regulates protein expression of wide number of targets through several mechanisms. It is conceivable that these mechanisms act in a target specific manner and that regulation of *Tnf* and *Nfkbid* also involves other Roquin-1 domains that are still present in the truncated variant.

Fig.21 Poly(A) tail analysis of *Icos*, *Tnf* and *Nfkbid* in 4-OHT treated Rc3h1-2^{fl/fl}; CD4Cre-ERT2; rtTA T cells transduced with GFP fused WT or R687* Roquin-1.

Fig. 7/S6: The reconstitution experiments of conditional Roquin1/2-deficient murine T-cells were well performed and controlled. These data are convincing, but in the end they do not uncover a role of R687* (the murine equivalent of R688*) for the regulation of major cytokines (TNF α , IL17A and IL-2) or OX40/ICOS implicated in the patient's disease (see Fig. 2F-K).

We agree with reviewer 4 that the data in Fig.7 do not provide direct prove for a role of the tested Roquin-1 targets in the development of the immune dysregulation observed in the R688*/R688* patient. Nonetheless, we want to stress that although the pathogenesis of hyperinflammatory syndromes is currently still a matter of debate, a central role of hyperactivated T cells has been proposed (Schulert et al. 2018). The analysis of the immune dysregulation in the R688*/R688* patient in the revised manuscript and the experiments performed for this rebuttal (Fig. 9,14,15 and 17) corroborate this hypothesis. This is further strengthened by the fact that the clinical signs of hyperinflammation in the patient waned upon CSA treatment. In the lights of these findings, the inability of the R687* Roquin-1 variant to control effector molecules of activated T cells further strengthens the concept that dysregulation of T cell activation is central in the disease pathogenesis of the R688*/R688* proband.

Comments reviewer 4:

General points:

1. Page 4, first paragraph. The word “symptoms” should be reserved for aspects of the disease that are subjectively felt by the patient, as opposed to signs.

The manuscript has been amended and changes have been highlighted in red.

2. P. 5, second paragraph. If the authors are going to use the Gene-/- terminology, they should use the correct gene name – if they are going to use the protein names, they should figure out another method of identifying the mutant or normal proteins.

The manuscript has been amended and changes have been highlighted in red.

3. Figure 2 is virtually impossible for a non-specialist to understand, at least as presented. It is not clear how many blood samples from the subject were analyzed, but it is possible that all the data described in this figure come from a single sample. My view is that a simple table showing enrichment or lack of enrichment of certain cell types would be easier to understand, along with a description of the number of samples analyzed for the subject with the mutation and some indication of the significance of the findings. Obviously, patient material may be limited, which is understandable, but it's important to convey how sure they are of these findings by detailing the numbers of replicates and, if possible, the significance of their differences with controls.

Reviewer 4 has raised similar concerns as Reviewer 3. To address these concerns, we have generated a revised Supp.Fig.2B (Fig.17 in this rebuttal). These figures/tables give more insight in the changes of the immune cell clusters on the one hand (Supp.Fig.2A) and promote the comparison with the traditional manual analysis on the other hand (Supp.Fig.2B).

Along similar lines, Supp. Fig.2B will allow the readers to observe that the observed phenotype of the patient is relatively stable given that replicating the data at different ages reveals a similar phenotype. This strengthens the data and we thank reviewer 4 for pointing this out.

Finally, we have restructured the result section to focus on the most significant changes in the immunophenotype of the R688*/R688* proband.

4. We appreciate the comparisons with the existing mouse models, but, based on the known domain structure, they are not perfect models of the human disease. I suspect the authors are trying to make the analogous mouse, but it might be helpful to have a short table or figure describing the differences in expected protein amount and sequence between the human point mutant and the available mouse mutants (and possibly the heterozygous Japanese subject), particularly in relation to the known domain structure of the protein.

A similar proposal has been made by Reviewer 1 and we kindly refer Reviewer 4 to Fig.6 of this rebuttal.

5. Ideally, the data on cytokine secretion and mRNA turnover would be performed on cells from the analogous point mutant mice, but since those are not available, the authors had to rely on overexpression studies in HEK 293 cells, and a complex retroviral replacement strategy in cells from mutant mice. In my view, they should at least note the potential limitations of these approaches, in contrast to studies of cells in mice and/or humans with the exact mutations under study in the endogenous protein.

We agree with Reviewer 4 concerning this particular limitation of using model systems to study human disease. The shortcomings of these approaches have been mentioned in the discussion section of the revised manuscript and are highlighted in red.

6. I may have missed this, but it would be of interest to know if the authors had searched the available public human databases for this variant, presumably in heterozygous form, to know if this variant has been detected in the general population, at least as reflected in these databases.

Indeed, up to now this specific variant has not been described in publicly available human databases such as Gnomad or the 1000 Genomes project. Of interest, somatic c.2062C>T mutations resulting in p.R688* have been identified in colon adenocarcinoma. Although briefly mentioned in the original manuscript, changes have been made in the revised manuscript and are highlighted in red.

Reviewers' Comments:

Reviewer #1:

Remarks to the Author:

The manuscript underwent substantial revision and modification. The manuscript has improved and the comments are appropriately addressed.

Reviewer #2:

Remarks to the Author:

The authors have comprehensively addressed critiques. I have no further comments or concerns.

Reviewer #3:

Remarks to the Author:

The authors have strengthened their conclusions by additional experiments (for example the inclusion of polyA tail length measurements in the presence/absence of wild type roquin or of the R687 mutant, Fig. 6H). They also provide a more balanced interpretation of their results. This includes a discussion of the evidence (from both the literature and experiments of their own study) concerning an additional contribution of the myeloid compartment to the clinical phenotypes of their patient. Altogether, these alterations along the lines suggested by the reviewers have significantly improved and focused this study.